# Dual transcranial electromagnetic stimulation of the precuneus boosts human long-term memory

Ilaria Borghi[1,2], Lucia Mencarelli[1], Michele Maiella[1], Elias Paolo Casula[1,3], Matteo Ferraresi[1], Francesca Candeo[1], Elena Savastano[1], Martina Assogna[1], Sonia Bonnì[1], Giacomo Koch[1,2]*

[1]Experimental Neuropsychophysiology Laboratory, Santa Lucia Foundation IRCCS, Rome, Italy; [2]Department of Neuroscience and Rehabilitation, University of Ferrara, and Center for Translational Neurophysiology of Speech and Communication (CTNSC), Italian Institute of Technology (IIT), Ferrara, Italy; [3]Department of System Medicine, "Tor Vergata" University of Rome, Rome, Italy

*For correspondence:
g.koch@hsantalucia.it

Competing interest: The authors declare that no competing interests exist.

## eLife Assessment

This work presents **important** findings suggesting that a combination of transcranial stimulation approaches applied for a short period could improve memory performance. **Solid** methods and evidence, in line with current standards for non-invasive stimulation and recording, are included to broadly support the main findings. The results potentially have implications for non-invasive enhancement of cognitive functions.

**Abstract** Non-invasive brain stimulation techniques have the potential to improve memory functions. However, the results so far have been relatively modest and time-consuming. Here, we implemented a novel 3-min combination of personalized repetitive transcranial magnetic stimulation (intermittent theta burst, iTBS) coupled with simultaneous application of gamma transcranial alternating current stimulation (γtACS) over the precuneus, a brain area connected with the hippocampus, to modulate long-term memory in healthy subjects. Only dual electromagnetic stimulation of the precuneus produced an increase in long-term associative memory as compared to iTBS alone and sham conditions in a sample of healthy volunteers. The effects were replicated in another independent sample, in which the increased associative memory was retained for up to 1 week. Moreover, dual stimulation increased gamma oscillations and precuneus–hippocampus functional connectivity through the white matter tracts linking the precuneus with the temporal lobe. These findings show that dual stimulation may lead neuronal assemblies in a state favorable to enhance long-term plasticity. Personalized dual electromagnetic stimulation of the precuneus may represent a new powerful approach for enhancing memory functions in several healthy and clinical conditions.

## Introduction

As the world's population ages, the prevalence of age-related memory deficits is growing. This phenomenon often signals the onset of more severe cognitive decline, which has a strong impact on society. In light of this escalating challenge, there is an urgent need for innovative strategies aimed at enhancing cognitive functions and potentially mitigating cognitive decline.

Non-invasive brain stimulation (NIBS) techniques have been largely used to enhance human cognition in the last two decades (*Antal et al., 2022*). Encouraging results have been reported in both

healthy and pathological conditions, including depression, stroke, and Alzheimer's disease (*Blumberger et al., 2018*; *Koch et al., 2022*; *Lefaucheur et al., 2020*).

Repetitive transcranial magnetic stimulation (rTMS) and transcranial alternating current stimulation (tACS) are two forms of NIBS widely used to enhance memory performance (*Grover et al., 2022*; *Koch et al., 2018*; *Wang et al., 2014*). rTMS, based on the principle of Faraday, induces depolarization of cortical neuronal assemblies and leads to after-effects that have been linked to changes in synaptic plasticity involving mechanisms of long-term potentiation (LTP) (*Huang et al., 2017*; *Jannati et al., 2023*). On the other hand, tACS causes rhythmic fluctuations in neuronal membrane potentials, which can bias spike timing, leading to an entrainment of the neural activity (*Wischnewski et al., 2023*). In particular, the induction of gamma oscillatory activity has been proposed to play an important role in a type of LTP known as spike timing-dependent plasticity, which depends on a precise temporal delay between the firing of a presynaptic and a postsynaptic neuron (*Griffiths and Jensen, 2023*). Both LTP and gamma oscillations have a strong link with memory processes such as encoding (*Bliss and Collingridge, 1993*; *Griffiths and Jensen, 2023*; *Rossi et al., 2001*), pointing to rTMS and tACS as good candidates for memory enhancement.

However, despite the increasing expectations, their overall effects are relatively modest, even with prolonged stimulation sessions (*Grover et al., 2022*; *Polanía et al., 2018*; *Wang et al., 2014*). In fact, the behavioral and clinical effects are extremely variable (*Corp et al., 2020*), especially when used to modulate memory functions (*Pabst et al., 2022*).

Recent studies explored the possibility of combining these two techniques, showing that the simultaneous application of tACS in the range of the gamma frequency band (γtACS) with intermittent theta burst (iTBS), an rTMS protocol known to induce LTP (*Huang et al., 2005*), enhances the after-effects of iTBS alone on the motor and prefrontal cortices (*Guerra et al., 2018*; *Maiella et al., 2022*).

Long-term memory formation has been associated with the neural activity of the precuneus (PC). The PC represents one of the main hubs of the default mode network (*Cavanna and Trimble, 2006*; *Cunningham et al., 2017*; *Jitsuishi and Yamaguchi, 2021*) and is involved in the formation of associative, episodic, and autobiographic memories (*Bonnì et al., 2015*; *Brodt et al., 2018*; *Flanagin et al., 2023*). Moreover, it is connected with the temporal lobe and the hippocampus (*Cunningham et al., 2017*; *Dadario and Sughrue, 2023*; *Tanglay et al., 2022*). Hence, the PC represents an ideal target for NIBS to modulate memory functions with profound clinical implications, such as in the case of Alzheimer's disease (*Koch et al., 2025*; *Koch et al., 2024*; *Koch et al., 2022*; *Koch et al., 2018*).

Here, we hypothesized that γtACS would activate key neural mechanisms underlying memory formation, favoring the iTBS-mediated plasticity induction by entraining and synchronizing endogenous gamma oscillations, thereby improving long-term memory. Thus, we reasoned that the dual application of iTBS and γtACS over the PC could be a promising tool for memory enhancement by enhancing plasticity mechanisms.

## Results

### Dual precuneus iTBS + γtACS improves long-term associative memory performance

In the first experiment, subjects (*n* = 20) were asked to perform two tasks measuring long- and short-term memory performances immediately after three balanced, randomized, and double-blind stimulation conditions with a cross-over design (iTBS + γtACS, iTBS + sham-tACS, sham-iTBS + sham-tACS). Stimulation parameters were personalized and identified using functional MRI (fMRI) and TMS–EEG (see *Figure 1A, B*). Long-term memory was assessed by the face–name associative task (FNAT), requiring the memorization of 12 faces with corresponding names and occupations, while short-term memory was assessed by the short-term memory binding test (STMB) (see *Figure 1C*).

The different stimulation protocols were all well-tolerated. No one reported significant side effects connected with the neuromodulation protocol applications. We found that dual iTBS + γtACS improved long-term associative memory as compared to the other conditions. Dual iTBS + γtACS increased the performance in recalling the association between face, name, and occupation (FNAT accuracy), both for the immediate ($F_{2,38}$ = 7.18; p = 0.002; $\eta^2_p$ = 0.274) and the delayed ($F_{2,38}$ = 5.86; p = 0.006; $\eta^2_p$ = 0.236) recall performances (*Figure 2*, panel A). The sensitivity analysis showed a minimal detectable effect size of $\eta^2_p$ = 0.215 with 20 participants. Post hoc analysis showed an advantage for dual iTBS

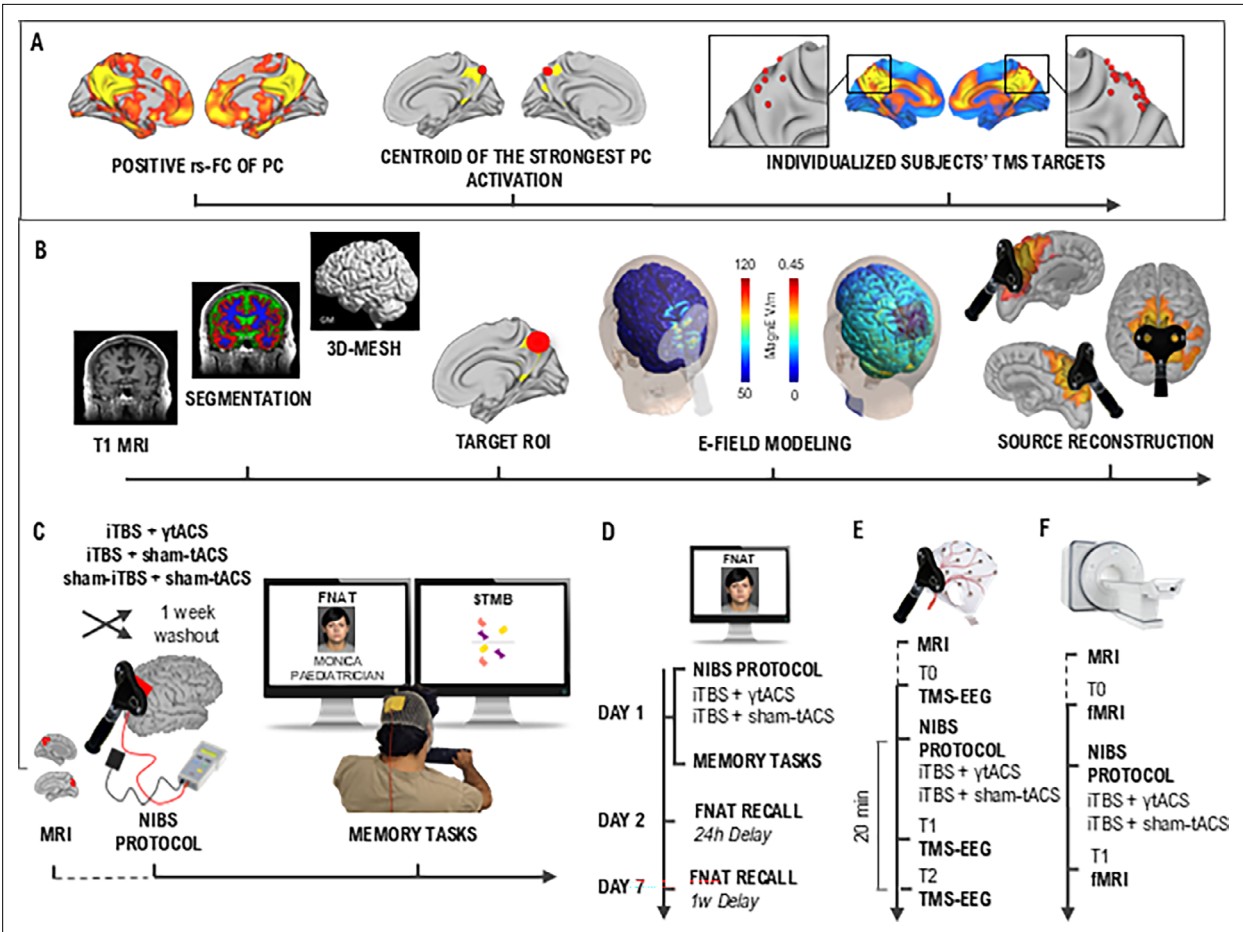

**Figure 1.** Individual target extraction and experimental design. (**A**) Methodological workflow used to extract the individualized target for the neuromodulation protocol used in experiments 1, 3, and 4. Participants underwent functional MRI (fMRI) scanning to individualize the stimulation sites and permit neuronavigation. Target individualization was derived by computing a PC functional connectivity profile for each participant, thus obtaining a map of positively correlated voxels, respectively, representing the DMN (panel A, left). The individual stimulation targets were defined as the centroid of the strongest PC activation being on the top of a cortical gyrus and representing the shortest perpendicular path connecting the stimulating TMS coil on the scalp and the cortex (panel A, center). PC coordinates for each participant are represented in red on an MNI brain template, showing the overlap with the DMN (panel A, right). (**B**) Biophysical modeling was computed for each subject, acquiring T1w and T2w MRI and using the SimNIBS toolbox for the T1w segmentation and the 3D-mesh transformation (panel B, left). The mean Norm e-field was extracted from a target region of interest (ROI) sphere (10 mm radius) centered on the individual coordinates of the PC (panel B, center). The simulated induced electric field is shown for a representative subject produced by iTBS (e-field modeling, left) and transcranial alternating current stimulation (tACS; e-field modeling, right). EEG source activity reconstruction induced by the TMS pulse over the precuneus in a representative subject (panel B, right). Experimental design of experiments 1 (**C**), 2 (**D**), 3 (**E**), and 4 (**F**). The effect of simultaneous iTBS + γtACS on memory performance was investigated in experiments 1 (**C**) and 2 (**D**) through two memory tasks: the face–name associative task (FNAT), which required the memorization of 12 faces with corresponding names and occupations, and the visual short-term memory binding test (STMB), which consisted of a change detection task. In the main experiment 1 (**C**), subjects were involved in a cross-over design with different experimental sessions of neuromodulation separated by a washout week. Every session corresponded to a different balanced and randomized stimulation condition (i.e., iTBS + γtACS, iTBS + sham-tACS, sham-iTBS + sham-γtACS) immediately followed by the FNAT learning phase and immediate recall, the STMB and the FNAT delayed recall (15 min delayed) and recognition. In experiment 2 (**D**), subjects were involved in a cross-over design with two balanced and randomized stimulation conditions (i.e., iTBS + γtACS, iTBS + sham-tACS) separated by a washout week. During the first session (day 1), participants received the neuromodulation protocol and then performed the learning phase and immediate recall FNAT, the STMB, and the FNAT delayed recall. In the second session (day 2), participants performed FNAT recall with a 24-hr delay from the neuromodulation protocol, while in the third session (day 7), participants performed FNAT recall and recognition with a 1-week delay. In experiment 3 (**E**), participants were involved in two randomized and balanced experimental sessions of neuromodulation (i.e., iTBS + γtACS, iTBS + sham-tACS) separated by a washout week. TMS–EEG recordings were performed before (T0), immediately after (T1), and 20 min after the neuromodulation (T2). In experiment 4 (**D**), after the first MRI scanning used for neuronavigation, participants were involved in two randomized and balanced experimental sessions of neuromodulation (i.e., iTBS + γtACS, iTBS + sham-tACS) separated by a washout week. The fMRI scanning was performed before (T0) and immediately after (T1) the neuromodulation protocol. Photographs reported represent the author Michele Maiella performing the task and an example of the face item used in the task taken from the FACES database (*Ebner et al., 2010*).

*Figure 1 continued on next page*

The online version of this article includes the following figure supplement(s) for figure 1:

**Figure supplement 1.** TMS and transcranial alternating current stimulation (tACS) e-field simulation.

**Figure supplement 2.** Face–name associative task (FNAT).

**Figure supplement 3.** Short-term memory binding task (STMB).

+ γtACS as compared to iTBS + sham-tACS condition for the immediate (40.0 ± 21.6% vs. 25.0 ± 17.9%, p = 0.002 Bonferroni corrected, corresponding to a relative improvement of 63.79%) and for the delayed recall (34.2 ± 20.4% vs. 24.2 ± 19.8%, p < 0.05 Bonferroni corrected, corresponding to a relative improvement of 34.16%), as well as compared to the sham-iTBS + sham-tACS condition for the delayed recall (34.2 ± 20.4% vs. 26.3 ± 17.6%, p=0.037 Bonferroni corrected, corresponding to a relative improvement of 34.04%) (*Figure 2A*).

The in-depth analysis of the FNAT accuracy investigating the specific contribution of face–name and face–occupation recall revealed that dual iTBS + γtACS increased the performance in the association between face and name (FNAT NAME) delayed recall ($F_{2,38}$ = 3.46; p = 0.042; $\eta^2_p$ = 0.154; iTBS + γtACS vs. sham-iTBS + sham-tACS: 42.9 ± 21.5% vs. 33.8 ± 19%; p = 0.048 Bonferroni corrected) (*Figure 2—figure supplement 1*). iTBS + sham-tACS did not result in any improvement as compared to sham-iTBS + sham-tACS. No significant effects were found over the recall of the association between face and occupation (FNAT OCCUPATION) (p < 0.05). Moreover, we found that the effects of dual iTBS + γtACS stimulation were selective for long-term but not short-term memory, since no effects were found on STMB accuracy and RTs (all ps >0.05) (*Figure 2B*). *Supplementary file 1* shows experiment 1 statistical details. Biophysical modeling calculations showed that the iTBS induced a mean e-field of 29.55 ± 5.17 V/m, while γtACS led to a much smaller mean e-field of 0.1 ± 0.55 V/m.

## Enhanced memory performance is still evident after 1 week

To confirm the above-described effects of dual iTBS + γtACS on an independent sample (n = 10) and to investigate the time course of memory retention, we performed a second experiment. In this experiment, subjects were asked to execute the delayed FNAT cued recall 24 hr and 1 week after the neuromodulatory protocols. As for experiment 1, we confirmed that dual iTBS + γtACS improved immediate recall at TOTAL FNAT as compared to iTBS + sham-tACS ($F_{1,9}$ = 7.31; p = 0.024; $\eta^2_p$ = 0.448; 26.7 ± 10.2% vs. 17.5 ± 8.3%; p = 0.024 corresponding to a relative improvement of 82.50%) (see *Supplementary file 2* for statistical details). The sensitivity analysis showed a minimal detectable effect size of $\eta^2_p$ = 0.388 with 10 participants. More importantly, we found that dual iTBS + γtACS exerted a long-lasting effect on long-term associative memory ($F_{1,9}$ = 8.433; p = 0.017; $\eta^2_p$=0.484) (*Figure 2C*). The sensitivity analysis showed a minimal detectable effect size of $\eta^2_p$=0.125 with 10 participants. We observed that the incremental effect on TOTAL FNAT performance was still evident at 24 hr and 1 week after the neuromodulation (day 1: 25.8 ± 10.7% vs. 15 ± 9.5%, $t_9$ = −2.75, p = 0.011 corresponding to a relative improvement of 78.33%; day 2: 25 ± 9.6% vs. 14.2 ± 9.7%, $t_9$ = −2.89, p = 0.009 corresponding to a relative improvement of 88.33%; day 7: 23.3 ± 12.9% vs. 14.2 ± 12.5%, $t_9$ = −2.40, p = 0.020 corresponding to a relative improvement of 80%) (see *Supplementary file 3*). No significant effects were found for STMB accuracy and RTs (all ps > 0.05) as for experiment 1 (see the *Supplementary file 2* for statistical details).

## Precuneus iTBS + γtACS increases gamma oscillatory activity

We then aimed to investigate the effects of the neuromodulation protocol on cortical oscillations (TMS-related spectral perturbation, TRSP) and excitability (TMS-evoked potentials, TEPs), through TMS–EEG recordings (n = 14 subjects, 6 of those also participated in experiment 1). We found that cortical oscillations in the gamma band increased after dual iTBS + γtACS as compared to baseline TMS–EEG recordings (stimulation $F_{1,13}$ = 5.073; p = 0.042; $\eta^2_p$ = 0.281) (*Figure 3A*). Specifically, we observed an increase in the gamma-TRSP after dual iTBS + γtACS as compared to iTBS +sham-tACS in both time points (ΔT1: 0.045 ± 0.09 to −0.013 ± 0.06; $t_{13}$ = 1.96; p = 0.036; ΔT2: 0.026 ± 0.04 to −0.013 ± 0.05; $t_{13}$ = 1.92; p = 0.039) (*Figure 3B*). This effect was specific to PC; indeed, no significant

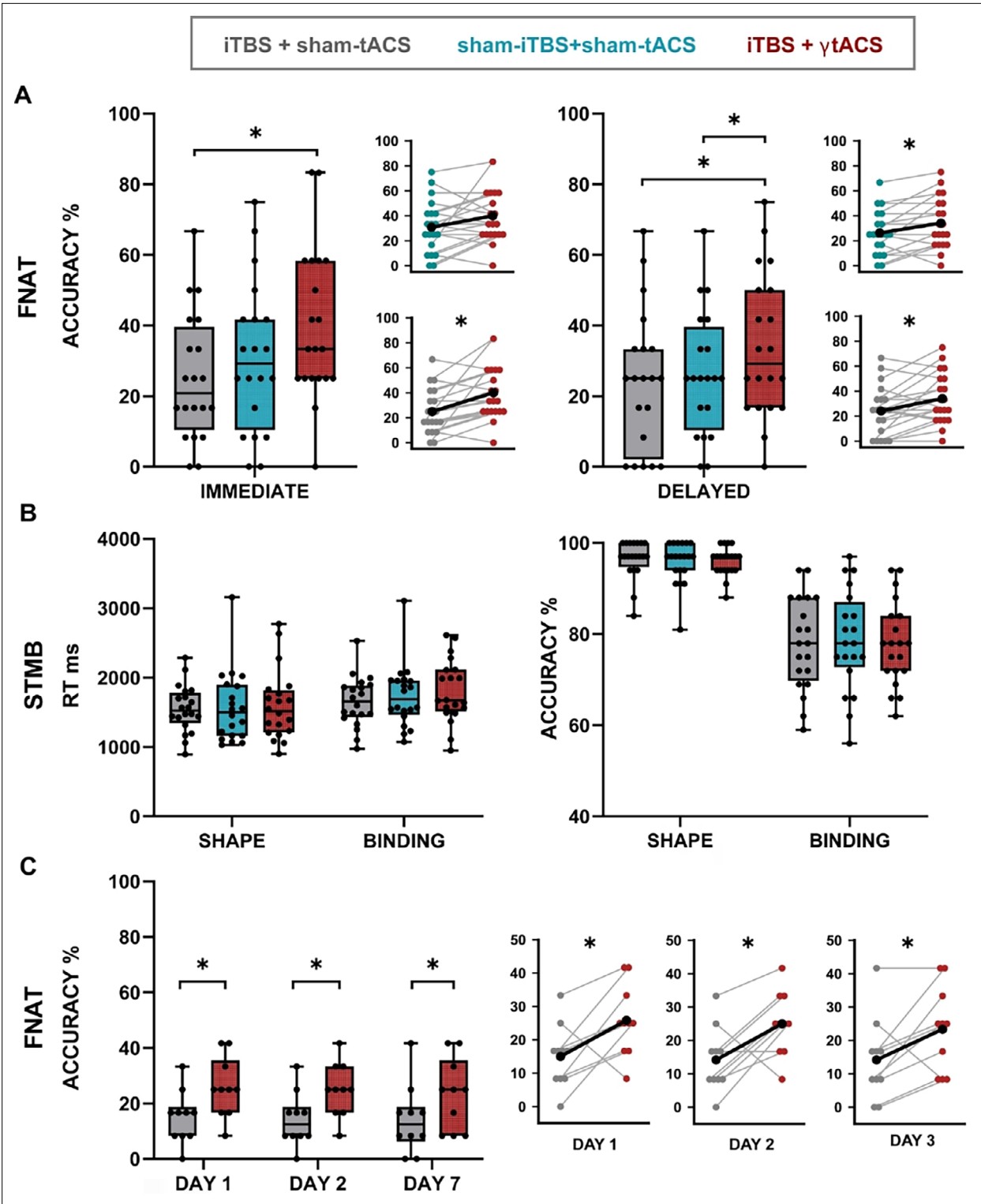

**Figure 2.** Memory performance outcome. The figure shows the results of the repeated measures ANOVA performed to analyze memory performance outcomes. In the Box and Whiskers plots, boxes delimit the lower (Q1) and the upper (Q3) quartiles, the whiskers extend to the smallest and largest values, and dots represent individual values. In the line graphs, colored dots show individual participant data, lines show the connection between the performances of each participant, and black dots and lines correspond to the mean performance values. Gray corresponds to iTBS + sham-tACS, blue to sham-iTBS + sham-tACS and red to iTBS + γtACS. *p < 0.05. (**A**) Face–name associative task (FNAT) accuracy in immediate (right) and delayed (left) trials resulting from experiment 1. *N* = 20. (**B**) Short-term memory binding task RTs (left) and accuracy (right) resulting from experiment 1. *N* = 20. (**C**) FNAT's long-lasting effect resulted from experiment 2. The results are shown over time (days 1, 2, and 7). *N* = 10.

*Figure 2 continued on next page*

*Figure 2 continued*

The online version of this article includes the following figure supplement(s) for figure 2:

**Figure supplement 1.** Face–name associative task (FNAT) in-depth analysis.

effect was found when testing TMS–EEG over the left posterior parietal cortex (l-PPC) ($p > 0.05$). This result was reflected by increased cortical excitability evident after dual iTBS + γtACS but not after iTBS + sham-tACS condition (*Figure 3C*). Specifically, we observed a significant TEP amplitude increase over a cluster of parietal and occipital electrodes comparing T0 vs. T1 (mean $t\text{-value}_{13} = -3.29$; all $ps < 0.05$) and over POz ($t\text{-value}_{13} = -4.74$; $p < 0.05$) when comparing T0 vs. T2. No significant differences were found comparing T0 vs. T1 and T2 in the iTBS + sham-tACS condition ($ps > 0.05$).

## iTBS + γtACS increases precuneus connectivity with the hippocampus

Finally, we investigated the effects of the dual iTBS + γtACS protocols on MRI-based resting state functional connectivity ($n = 16$ subjects, 7 of whom participated in experiment 1). Region of interest (ROI)-to-ROI analysis revealed a significant difference specifically for the connectivity between the

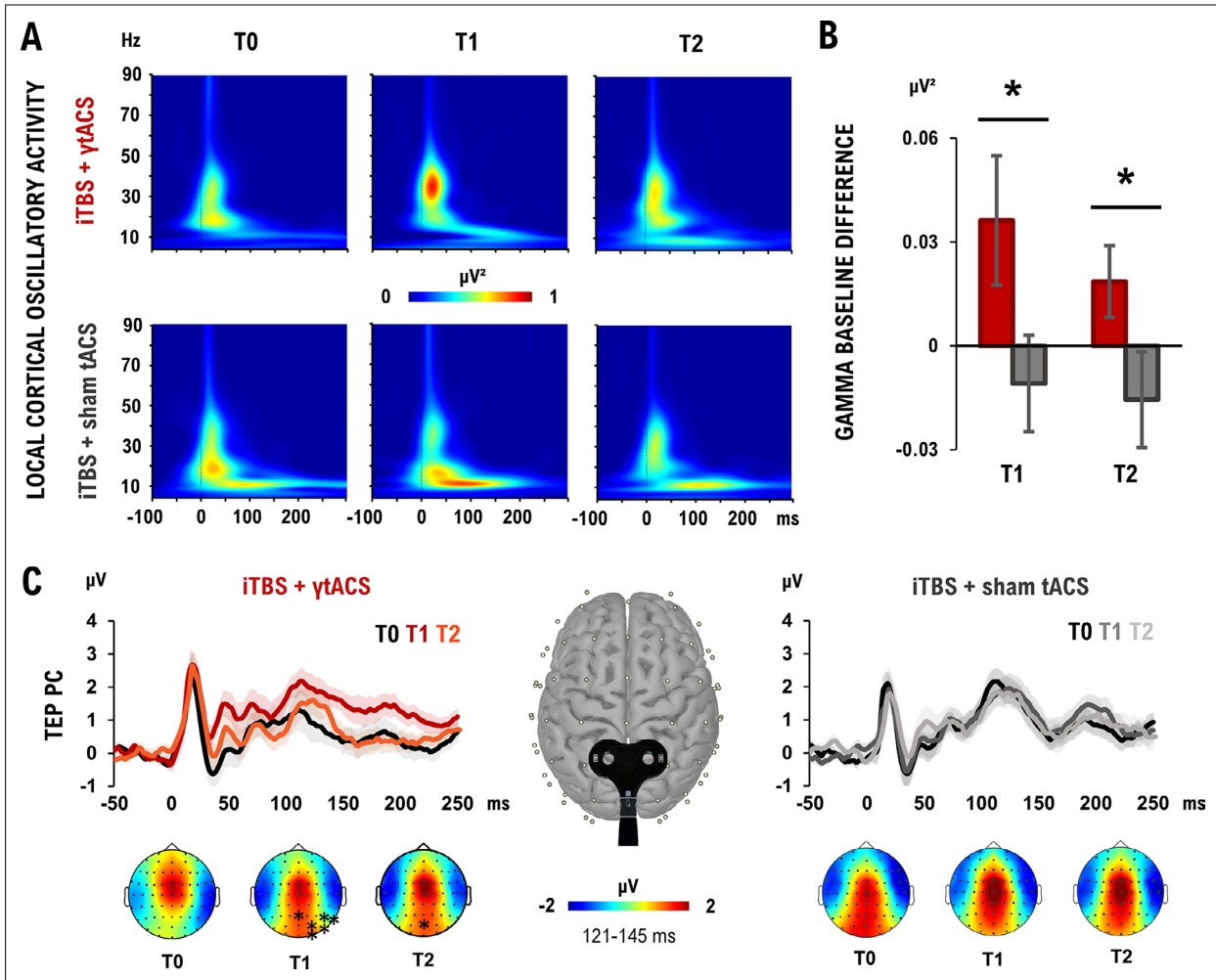

**Figure 3.** Experiment 3 neurophysiological outcome. (**A**) Precuneus (PC) oscillatory activity elicited by iTBS + γtACS (up) and iTBS + sham-tACS (down) when testing PC over the three time points (T0, T1, T2 from left to right) with repeated measures ANOVA. (**B**) Gamma oscillation changes from baseline after iTBS + γtACS (red) and iTBS + sham-tACS (gray). (**C**) TMS-evoked potential (TEP) produced over the PC when performing TMS–EEG over PC in the two stimulation conditions: iTBS+γtACS (up-left) and iTBS + sham-tACS (up-right) over the three time points (T0, T1, T2). (Down) Topographies and statistical differences in TEPs amplitude after the different stimulation conditions (iTBS + γtACS, left; iTBS + sham-tACS, right) over the three time points (T0, T1, T2, from left to right). $N = 14$; *$p < 0.05$; bars depict standard error.

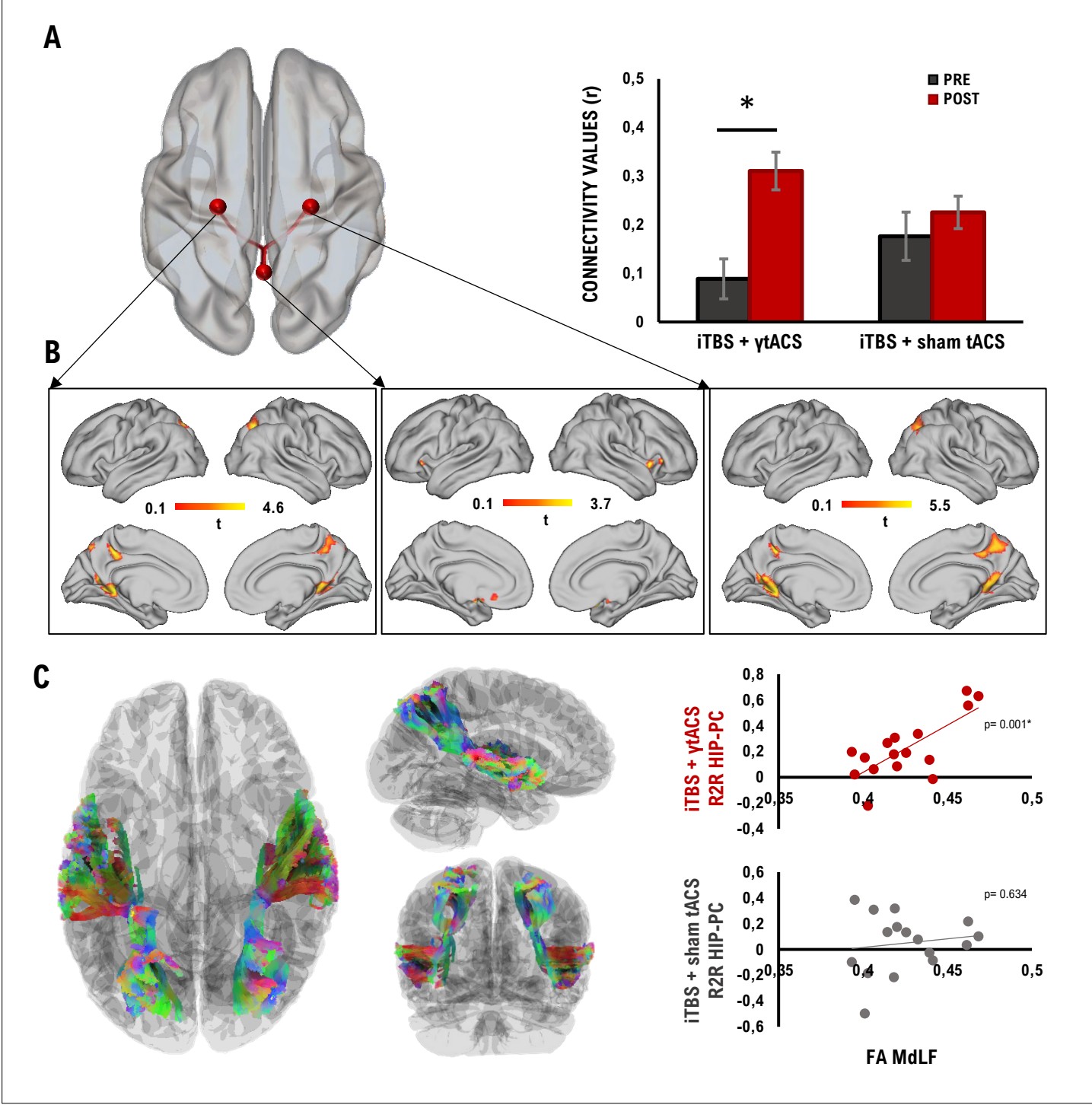

**Figure 4.** Resting-state functional connectivity (rs-FC) changes after iTBS + γtACS and correlation with Diffusion Tensor Images (DTI). (**A**) The standard MNI brain (left) and the bar plot (right) display the positive correlation between the PC and the bilateral HIP after the iTBS + γtACS resulted from the region of interest (ROI)-to-ROI analysis. (**B**) Seed-to-voxel analysis results from each significant ROI (i.e., left HIP, left; PC, center; right HIP, right) are overlaid on a standard MNI brain. (**C**) Middle Longitudinal Fasciculus (MdLF) extracted (left); positive correlation between MdLF integrity and functional connectivity changes between the PC and bilateral HIP after iTBS + γtACS (upper right); the absence of correlation in the iTBS +sham-tACS condition and functional connectivity (lower right). *N* = 16; *p < 0.05; bars depict standard error.

PC and the hippocampi ($F_{3,13}$ = 5.78, p-FDR = 0.029). Post hoc analysis showed increased connectivity after the stimulation for the dual iTBS + γtACS ($t_{15}$ = 3.67, p-FDR = 0.0023; *Figure 4A*), but not for the iTBS + sham-tACS. No significant differences were detected among the conditions before stimulation (all p > 0.05). The significant nodes that emerged from the ROI-to-ROI analysis (i.e., PC and hippocampi) were considered for the seed-to-voxel analyses, revealing an increased resting-state functional connectivity (rs-FC) between the PC and the orbitofrontal cortex after dual iTBS + γtACS as compared to baseline OFC; MNI coordinates: x = 20, y = 16, z = −4; |T15| > 3.20, k = 1051; p = 0.0003 (*Figure 4B*, center). Moreover, stronger connectivity with the PC (MNI coordinates: x = 30, y = −70, z = 46; |T15| > 3.20, k = 1499; p = 0.000167) and with the posterior cingulate cortex emerged when considering the right hippocampus as seed (MNI coordinates: x = −14, y = −50, z = 12; |T15| > 3.20, k = 1295; p = 0.0001649) (*Figure 4B*, right). Finally, similar results occurred considering as seed the left hippocampus, with an improved rs-FC with the PC (MNI coordinates: x = 30, y = −68, z = 44; |T15| > 3.20, k = 1525; p = 0.000163) and with the posterior cingulate cortex (MNI coordinates: x = 20, y = −50, z = 12; |T15| > 3.20, k = 509; p = 0.022) (*Figure 4B*, left). We also reconstructed the white matter fibers connecting the PC with the temporal lobe in each subject, forming the bilateral Middle Longitudinal Fasciculus (MdLF), using Diffusion Tensor Images (DTI) tractography. We found that the structure of the MdLF measured through fractional anisotropy (FA) was related to the increased connectivity between PC and the hippocampi after dual iTBS + γtACS. A significant positive correlation emerged between the FA of the MdLF and the changes in connectivity between PC and hippocampus in the iTBS + γtACS (r = 0.73, p = 0.001), but not in the iTBS + sham-tACS (r = 0.14, p = 0.634) (*Figure 4C*).

## Discussion

Here, we demonstrated that the combination of iTBS and γtACS applied over the personalized PC can enhance long-term associative memory formation. Dual stimulation increased the ability to recall the association between faces, names, and occupations as compared to the iTBS alone or sham conditions. Moreover, the increased memory trace was long-lasting, being evident 1 week after the stimulation. On the other hand, iTBS alone did not result in any memory improvement, in agreement with previous studies showing contradictory effects of this protocol on neuroenhancement (*Hamada et al., 2013*; *Ziemann and Siebner, 2015*). Hence, we show that the combination of neuroplasticity induction promoted by iTBS and gamma oscillations entrainment produced by γtACS may be the key to memory boosting. We believe that the novel results reported here may have several implications for treating memory impairments, gamma dysregulation, and network dysfunction in a large variety of conditions.

Our results seem to identify the PC as having a role in mediating long-term memory formation. Given the existing evidence on TMS propagation and the computation of the biophysical model with the e-field, we can assume that the individually identified PC was involved in the observed effects (*Ridding and Rothwell, 2007*). Moreover, we observed specific cortical changes in the posteromedial parietal areas, as evidenced by the whole-brain analysis conducted on TMS–EEG data and the absence of effect on the lateral posterior parietal cortex used as a control condition. The PC has been associated with memory engram formation and retrieval (*Brodt et al., 2018*; *Flanagin et al., 2023*; *Hebscher et al., 2019*). Moreover, it is a prime candidate in processing relations among entities, including semantic concepts, making its activity fundamental for associative memory formation (*Fernandino et al., 2022*; *Rentz et al., 2011*; *Summerfield et al., 2020*). These notions found support in our current findings, showing that neuromodulation of the PC resulted in an improvement of long-term associative memory.

Visual short-term associative memory, measured by STBM performance, was not modulated by any experimental condition. Even if we cannot exclude the possibility that the stimulation could have influenced short-term verbal associative memory, we expected this result since short-term associative memory is known to rely on a distinct frontoparietal network while FNAT, used to investigate long-term associative memory, has already been associated with the neural activity of the PC and the hippocampus (*Parra et al., 2014*; *Rentz et al., 2011*). Moreover, we observed a main effect on the ability to concomitantly recall face, name, and occupation, while for single association, we observed minimal (face–name) or no effect (face–occupation). Forming face–name associations is widely acknowledged to be particularly difficult due to the lack of contextual properties to formulate an associative link (*Werheid and Clare, 2007*). In contrast, forming an association between a face and

other biographical information, such as occupations or hobbies, is easier (*McWeeny et al., 1987*). We can hypothesize that retaining multiple associations (face–name–occupation) together is even more effortful and cognitively demanding. For these reasons, the stimulation could have promoted effects in a gradient way related to the difficulty of the task.

Apart from improving long-term memory, dual iTBS + γtACS promoted a long-term increase in gamma oscillatory activity, as measured by TMS–EEG. Cortical gamma-band oscillations are thought to be primarily generated by parvalbumin-positive GABA-ergic interneurons interacting with excitatory pyramidal cells (*Buzsáki and Wang, 2012*; *Cobb et al., 1995*). The repeated activation of these circuits through gamma oscillations fosters LTP formation, which is the neurobiological basis of long-term memory (*Bikbaev and Manahan-Vaughan, 2008*; *Debanne and Inglebert, 2023*; *Headley and Weinberger, 2011*). This finding is consistent with the idea that increased gamma oscillatory activity may facilitate cortical plasticity, neural communication, and mechanisms of memory encoding and retrieval, thereby ensuring the successful formation of an episodic memory (*Griffiths and Jensen, 2023*). We hypothesize that γtACS, by entraining and synchronizing ongoing oscillatory gamma activity, may have facilitated the formation of LTP induced by iTBS.

Furthermore, we found that dual iTBS + γtACS enhanced the connections between the PC and the hippocampus as measured by fMRI connectivity analysis. Effective neural communication driven by gamma oscillations ensures that, during memory formation, incoming information processed in the neocortex activates the relevant cell assemblies in the hippocampus to ensure associative binding (*Griffiths and Jensen, 2023*). Gamma-band oscillatory activity drives network connectivity, allowing for information processing and binding through synchronization between distant cortical areas (*Fries, 2005*; *Jensen et al., 2007*; *Miltner et al., 1999*; *Osipova et al., 2006*). Hence, we argue that PC gamma oscillations could have bound relevant stimuli for perceptual representation, while synchronization between cortical and hippocampal neurons could have allowed the representation to be encoded into the hippocampus (*Nyhus and Curran, 2010*).

Finally, our findings seem to suggest that higher functional connectivity was supported by higher white matter integrity (FA) of MdLF, the main white matter tract connecting the PC with the temporal lobe. Altogether, these results help to trace the possible anatomo-functional network engaged by PC stimulation, providing further support for the successful involvement of long-range pathways connecting the PC with the temporal lobe underlying memory formation.

A strength of the current work is the personalization of stimulation parameters that was achieved by targeting the PC spot more connected with the temporal lobe and by selecting the intensity of stimulation for iTBS using TMS–EEG. Hence, we argue that personalization of NIBS protocols is a key factor in reducing the inter-individual variability that currently limits the use of common protocols such as TBS and in increasing the overall beneficial effects on cognitive functions.

Why should the dual application of γtACS + iTBS lead to such a strong boosting effect as compared to iTBS alone? It is well established that tACS alone does not induce relevant after-effects, especially when applied for a short period of 3 min, such as in the current case (*Guerra et al., 2019*; *Guerra et al., 2018*). However, during stimulation, tACS alters membrane potential toward depolarization or hyperpolarization in an oscillatory fashion, manipulating the excitability of neurons that become aligned with the introduced electric field, mostly pyramidal cells in layer V (*Fröhlich and McCormick, 2010*; *Guerra et al., 2018*; *Reato et al., 2013*; *Schutter and Hortensius, 2011*). Notably, these neurons are characterized by intrinsic resonance and neuroplastic activity. In particular, synchronous neural activation in the γ range is considered crucial for the induction of neuroplasticity and memory formation (*Buzsáki and Wang, 2012*). Gamma oscillations can synchronize the firing of multiple presynaptic neurons so that they exert a stronger depolarizing effect on the target postsynaptic neuron than if they were to fire in isolation (*Sjöström et al., 2001*). Synchronizing multiple inputs to a postsynaptic neuron enhances the likelihood of LTP. Hence, gamma oscillations are perhaps ideal because they provide a comparatively short window of excitability that ensures all neurons fire in near-perfect unison (*Jensen et al., 2007*). This provides a long-sought link between gamma oscillations, LTP, and the formation of new memories. For these reasons, γtACS may have provided the ideal neural substrate for the iTBS to exert its plasticity induction mechanisms. This might have occurred by driving neuronal assemblies in a state devoted to inducing long-term changes in cortical networks involved in memory formation. In humans, iTBS has been inspired by protocols used in animal models to elicit LTP in the hippocampus (*Huang et al., 2005*).

On top of iTBS, γtACS may have synchronized PC neuronal elements, boosting gamma oscillatory activity during the induction of LTP driven by iTBS. Notably, this interaction is rather specific, since previous studies showed that the application of γtACS alone does not induce any after-effects, while only gamma, but not theta or beta tACS, combined with iTBS induces long-lasting changes in cortical excitability (*Guerra et al., 2018*; *Maiella et al., 2022*). Moreover, our biophysical modeling calculations showed that γtACS induced a marginal augmentation of the e-field (~0.15 mV), which is below the amount of induced current needed to induce an effect in superficial cortical neurons (*Opitz et al., 2016*). Hence, tACS alone cannot account for the observed increase in memory as well as for enhancing cortical activity.

This study has some limitations. Although we studied TMS and tACS propagation through the e-field modeling and observed an increase in the precuneus gamma oscillatory activity, excitability, and connectivity with the hippocampi, we cannot exclude that our results might reflect the consequences of stimulating more superficial parietal regions other than the precuneus, nor report direct evidence of microscopic changes in the brain after the stimulation. Invasive neurophysiological recordings in humans for this type of study are not feasible due to ethical constraints. Studies on cadavers or rodents would not fully resolve our question due to significant differences between them (i.e., rodents do not have an anatomical correspondence, while cadavers have alterations in electrical conductivity occurring in postmortem tissue). However, further exploration of this aspect in future studies would help in the understanding of γtACS + iTBS effects.

We did not assess the effects of γtACS alone. This decision was based on the findings of *Guerra et al., 2018*, who investigated the same protocol and reported no aftereffects. Given the substantial burden of the experimental design on patients and our primary goal of demonstrating an enhancement of effects compared to the standalone iTBS protocol, we decided to leave out this condition. While examining the effects of γtACS alone could help isolate its specific contribution to this target and memory function, extensive research has shown that achieving a cognitive enhancement aftereffect with tACS alone typically requires around 20–25 min of stimulation (*Grover et al., 2023*). We did not study memory functions, gamma oscillations, and synchronization between the PC and hippocampus in the same session due to technical limitations with the current techniques (i.e., fMRI and TMS–EEG). Consequently, these findings do not allow precise inferences regarding the specific mechanisms by which dual iTBS and γtACS of the precuneus modulate learning and memory. Methodological restraints related to the high-frequency protocols employed in this study did not allow for closed-loop implementation. Moreover, we did not test tACS applied at different frequency bands combined with iTBS since our working hypothesis was based on the combination of gamma activity and plasticity induction.

Future studies should further investigate the effects of stimulation on distinct memory processes. In particular, stimulation could be applied before retrieval (*Rossi et al., 2001*) to better elucidate its specific contribution to the observed enhancements in memory performance. Additionally, it would be worth examining whether repeated stimulation, administered both before encoding and before retrieval, could produce a boosting effect. This is especially relevant in light of findings showing that matching the brain state between retrieval and encoding can significantly enhance memory performance (*Javadi et al., 2017*).

Our findings may have profound social and clinical implications.

These results may assume relevance in the context of memory impairment, which is an increasing and challenging aspect of the elderly population. The current findings may be relevant in the case of Alzheimer's disease, where the progressive memory deficit is accompanied by early involvement of DMN with an accumulation of beta-amyloid plaques, neurofibrillary tangles, atrophy, and functional connectivity primarily targeting the PC (*Billette et al., 2022*; *Buckner et al., 2005*; *Chen et al., 2017*; *Raichle et al., 2001*). In this perspective, the DMN has been recently identified as a new potential therapeutic target for neuromodulation in AD (*Koch et al., 2025*; *Koch et al., 2024*; *Koch et al., 2022*; *Koch et al., 2018*). Moreover, by linking together items and concepts, long-term associative memory is essential for the construction of declarative memory and, broadly, for learning and everyday functioning. Hence, dual magnetic and electrical transcranial stimulation could be promising in several social and clinical contexts, such as learning deficits, autism spectrum, and attention deficit hyperactivity disorders (*Mathalon and Sohal, 2015*).

## Materials and methods

### Experimental design

We conducted four experiments aimed at understanding the effects of dual iTBS + γtACS on associative memory performance, cortical oscillatory activity and reactivity, and functional connectivity.

#### Experiment 1

We investigated the effect of dual iTBS + γtACS on memory performance through two memory tasks: the FNAT and the visual STMB. Participants first underwent MRI scanning to individualize the stimulation sites and permit neuronavigation (*Figure 1A*), then they were involved in a double-blind cross-over design with three experimental sessions of neuromodulation separated by a washout week (*Figure 1C*). Every session corresponded to a different balanced and randomized stimulation condition (i.e., iTBS + γtACS, iTBS + sham-tACS, sham-iTBS + sham-γtACS) immediately followed by the two memory tasks. γtACS alone was not administered since a single 3-min γtACS does not exert any relevant after-effect (*Guerra et al., 2018*).

#### Experiment 2

We deepen the study of memory effects by confirming experiment 1 results and investigating the course of the memory trace over time after the neuromodulation protocol. We simplified the experimental design by excluding one of the control conditions, i.e., 'sham-iTBS + sham-γtACS', as we did not observe any effect related to this condition (see 'results' paragraph). In particular, participants were involved in a double-blind cross-over design with two balanced and randomized stimulation conditions (i.e., iTBS + γtACS, iTBS + sham-tACS) separated by a washout week. Participants were required to attend three experimental sessions for each condition. During the first session (day 1), participants received the neuromodulation protocol and then performed FNAT and STMB. In the second session (day 2), participants performed FNAT recall with a 24-hr delay from the neuromodulation protocol, while in the third session (day 7), participants performed FNAT recall and recognition with a 1 week delay (*Figure 1D*).

#### Experiment 3

We investigated the neurophysiological effects of the neuromodulation protocol through the combined use of TMS and electroencephalography (TMS–EEG). As for experiment 1, participants underwent an MRI scan to individualize the stimulation sites and permit neuronavigation. Then, they were involved in two balanced, double-blind, randomized experimental sessions of neuromodulation (i.e., iTBS + γtACS, iTBS + sham-tACS) separated by a washout week. TMS–EEG recordings were performed before (T0), immediately after (T1), and 20 min after the neuromodulation (T2) (*Figure 1E*).

#### Experiment 4

We investigated the effects of the neuromodulation protocol on functional connectivity through fMRI scanning. After the first MRI scanning used for neuronavigation, participants were involved in two balanced, double-blind, randomized experimental sessions of neuromodulation (i.e., iTBS + γtACS, iTBS + sham-tACS) separated by a washout week. The fMRI scanning was performed before (T0) and immediately after (T1) the neuromodulation protocol (*Figure 1F*).

### Participants

We recruited young healthy participants who provided written informed consent approved by the Santa Lucia Foundation IRCCS ethical committee (CE/PROG.923) in accordance with the Declaration of Helsinki. In total, 41 subjects were enrolled. Twenty-four participants were involved in experiment 1, in which we tested the effects of the dual iTBS and γtACS on memory functions. Another group of 10 was enrolled in experiment 2 to replicate the results of experiment 1 in an independent sample and to investigate the long-term effect of the dual stimulation. Sixteen participants were recruited in experiment 3, of which 10 had already participated in experiment 1, to examine the effects on cortical oscillatory activity and cortical reactivity. Finally, 18 participants were involved in experiment 4, of which 12 had already participated in experiment 1, to investigate functional connectivity through fMRI.

Eight participants (i.e., $n = 4$ from experiment 1, $n = 2$ from experiment 3, $n = 2$ from experiment 4) voluntarily withdrew before completing all the experimental sessions. We analyzed at all time points in $n = 20$ subjects for experiment 1, $n = 10$ for experiment 2, $n = 14$ for experiment 3, and $n = 16$ for experiment 4 (see *Supplementary file 4* for demographic characteristics).

## Enrollment criteria

Participants were recruited from the Rome metropolitan area via advertisements on local and electronic bulletin boards. Inclusion criteria were young age (18–40 years), native or fluent in Italian, and normal or corrected-to-normal vision and hearing. Exclusion criteria were left-handedness, claustrophobia, psychotropic medication, current or past psychiatric or neurological disorders, history of seizure, metal implant in the head, and implanted electronic device. TMS and tACS safety guidelines and medical regulations were fully followed (*Antal et al., 2017*; *Rossi et al., 2021*).

## Sample size estimation

The sample size for the main experiments 1, 3, and 4 was estimated based on a previous study in which we applied dual iTBS + tACS at different frequencies (theta and gamma) over the dorsolateral prefrontal cortex using TMS–EEG as a read-out of oscillatory cortical activity (*Maiella et al., 2022*). This study reported an effect size of 0.291 based on a two-way repeated-measures ANOVA in which we compared three neuromodulatory protocols (i.e., iTBS + γtACS; iTBS + sham-tACS; iTBS + theta-tACS at three different time points). This analysis revealed a specific increase in gamma activity after the iTBS-γtACS neuromodulatory protocol. Based on this effect size, our power analysis estimated that a sample size of 14 patients would be necessary to obtain the same effect size with 80% power and an alpha level of 0.05.

## iTBS + γtACS neuromodulation protocol

The co-stimulation consisted of a combination of iTBS and tACS delivered in the gamma band (γtACS) based on previous studies (*Guerra et al., 2018*; *Maiella et al., 2022*). The neuromodulation protocol was delivered on the individual PC, based on the individual resting state structural MRI and fMRI (see 'MRI data acquisition and preprocessing' paragraph), and targeted with a stereotaxic neuronavigation system (SofTaxic, E.M.S., Bologna s.r.l.). The active tACS electrode (anode) was placed on the scalp with the iTBS coil above it, and the other tACS electrode (cathode) was placed over the right shoulder's muscle (*Figure 1C*).

tACS was delivered through a Brainstim multifunctional system for low-intensity transcranial electrical stimulation (E.M.S., Bologna s.r.l.) and saline-soaked sponge electrodes ($7 \times 5$ cm$^2$). γtACS sinusoid frequency wave was set at 70 Hz with an intensity of 1 mA for a total duration of 190 s. iTBS was delivered through a MagStim Rapid$^2$ magnetic stimulator (Magstim Company, Whitland, Wales, UK) delivering a biphasic waveform pulse (pulse width ~0.1 ms) connected to a figure-of-eight coil (70 mm). iTBS consisted of ten bursts of three pulses at 50 Hz lasting 2 s, repeated every 10 s with an 8 s pause between consecutive trains, for a total of 600 pulses lasting 190 s (*Huang et al., 2005*). Biophysical modeling and e-field calculation were conducted to control for the e-field distribution. Stimulation parameters were personalized and identified using fMRI, TMS–EEG, and electromyography.

No ramp-up and no ramp-down were programmed for the stimulation. Sham stimulation conditions were implemented to control the individual contribution of the techniques and the placebo effect. tACS sham conditions were implemented, applying only a 2-s ramp up and 2-s ramp down at the beginning and at the end of stimulation, to give the participant the feeling of real stimulation. Sham-iTBS conditions were implemented by adding a wood layer under the coil (*Sandrini et al., 2020*). *iTBS intensity*. To select the intensity for the iTBS protocol, we first computed the resting motor threshold (RMT), defined as the lowest intensity producing MEPs of >50 μV in at least five out of 10 trials (*Rossini et al., 2015*). RMT was tested over the relaxed first dorsal interosseous (FDI) hotspot of the primary motor cortex (M1) in the left dominant hemisphere, with the tACS electrode under the coil to guarantee the same scalp-to-coil distance of the neuromodulation protocol. Electromyographic activity was recorded from the contralateral FDI muscle, using two Ag–AgCl surface cup electrodes (9 mm) in a belly-tendon montage. Responses were amplified through a Digitimer D360 amplifier (Digitimer Ltd, Welwyn Garden City, Hertfordshire, UK). Filters were set at 20 Hz and 2 kHz, with a sampling rate of 5 kHz. iTBS and tACS start times were synchronized using a BrainTrigger (E.M.S.,

Bologna s.r.l.) and SIGNAL Software. Since the coil-to-cortex distance directly influences the magnitude of magnetic stimulation, for each patient, we subsequently calculated a distance-adjusted RMT (AdjRMT). AdjRMT = RMT + m × (DsiteX – DM1) where AdjRMT is the adjusted MT in % of stimulator output, MT is the unadjusted MT in % of stimulator output, DM1 is the distance between the scalp and M1 hotspot, DSiteX is the distance between the scalp and a second cortical region (SiteX), and m is the distance-effect gradient (*Stokes et al., 2007*). Afterward, to optimize the stimulation intensity, each patient received 50 TMS single pulses at an initial intensity of 100% of adjMT over the individualized PC during a 64-channel EEG recording, which permitted the visualization of TEPs (*Mancuso et al., 2021*). The intensity of TMS was eventually increased in steps of 2% of the maximal stimulator output based on the visualization of a first TEP peak of at least 3 μV (*Koch et al., 2022*) reaching the effective stimulation intensity of PC (PC-eSI). Finally, the stimulation intensity for iTBS was set at 80% of the PC-eSI.

## Randomization and masking

Memory tasks, TMS–EEG, and fMRI were performed by researchers blinded to the neuromodulation condition (memory task: I.B., F.C., and E.S.; neurophysiological data: I.B. and M.M.; fMRI: L.M.). The neuromodulation protocols were performed by a dedicated technician (M.F.). All subjects received every stimulation condition thanks to the cross-over design. The stimulation conditions orders were randomized and balanced among subjects. Moreover, in experiments 1 and 2, the parallel forms of the memory task were equally distributed among the stimulation conditions orders.

## Biophysical modeling and e-field calculation

We calculated the norm e-field distribution induced by tACS and iTBS over the PC for each participant included in experiments 1, 3, and 4 using the simulation package SimNIBS (version 4.0) (*Figure 1*, panel B). The realistic head model was built from the structural T1- and T2-weighted MRI of each single subject. Default isotropic conductivities were used in our simulation (*Thielscher et al., 2015*). The final mesh was comprehensive of gray and white matter, scalp tissue, bone, and cerebrospinal fluid (see *Windhoff et al., 2013* for further modeling details). Each tissue segmentation was carefully examined slice-by-slice to ensure proper classification.

The TMS e-field distribution was computed using the model of the Magstim 70 mm figure-of-eight coil, used for the iTBS (*Thielscher and Kammer, 2002*). We calculated the e-field input in the form of d*I*/d*t* in units of A/μs based on the coil model, stimulator model, and pulse intensity for each subject (for details see *Kammer et al., 2001*). The center of the coil was positioned over the target area, expressed in individual MNI coordinates, as extracted from the baseline MRI. The coil model was then manually moved and rotated according to the specified position and direction used in the real stimulation. The tACS e-field distribution was calculated using the 7x5 rectangular sponge electrodes with an electrode thickness of 2 mm and a sponge thickness of 4 mm, centered in the target area of stimulation. Moreover, using an ad hoc script, we simulated the e-field distribution of the combined techniques (iTBS + tACS) for each subject.

The output is reported in *Supplementary file 5* and is represented in V/m. *Figure 1—figure supplement 2* also shows TMS and tACS e-field calculation for a representative subject.

## Memory tasks

### Face–name associative task

In experiments 1 and 2, just after the neuromodulation protocol, the participants underwent the FNAT, a cross-modal associative memory test that requires participants to pair pictures of unfamiliar faces with common first names and occupations (*Papp et al., 2014*).

The task consisted of a learning phase followed by a recall and recognition trial. During the learning phase, 12 faces associated with a name and an occupation were shown for 8 s on a computer screen. Participants were instructed to read aloud the name and the occupation, to ensure their attention was focused on the items, and to memorize the face–name–occupation association. Right after the learning phase, there was an immediate cued recall phase, in which the participants were asked to recall the name and the occupation of each face previously shown while observing the faces again for 8 s each. The delayed cued recall consisted of showing the same faces and asking participants to say each name and occupation. Finally, during the recognition trial, participants were asked to

recognize the studied face from a distractor face matched for age and sex. Then, for those names and/or occupations not recollected, they had to identify the correct name and/or occupation among three alternatives each. The distractors were a novel name/occupation and a name/occupation associated with a different face. We considered a correct association when a subject was able to recall all the information for each item (i.e., face, name, and occupation), resulting in a total of 36 items to learn and associate. To further investigate the effect on FNAT, we also computed a partial recall score accounting for those items where subjects correctly matched only names with faces (FNAT NAME) and only occupations with faces (FNAT OCCUPATION). In experiment 1, the learning phase was followed by an immediate cued recall and a 15-min delayed cued recall with recognition (*Figure 1—figure supplement 2A*), while in experiment 2, the learning phase was followed by an immediate cued recall and a 15-min delayed recall on day 1, a 24-hr delayed cued recall on day 2, and a 1 week delayed cued recall with recognition on day 7 (*Figure 1—figure supplement 2B*). The recognition trial was administered just in the third session to avoid a learning bias over day 2 and 7 cued recall.

For this study, FNAT was adapted from an Italian version to create three parallel forms by selecting images from an online dataset (i.e., FACES, Center for Lifespan Psychology, Max Planck Institute for Human Development, Berlin, Germany) and pairing them with Italian names and occupations (*De Simone et al., 2023*; *Ebner et al., 2010*). Each picture's face size was 19 × 15 cm. In the learning phase, faces were shown along with names and occupations for 8 s each (totaling approximately 2 min). During immediate recall, the faces were displayed alone for 8 s. In the delayed recall and recognition phase, pictures were presented until the subject provided answers. We used a different set of stimuli for each stimulation condition, resulting in a total of three parallel task forms balanced across conditions and session order. All parallel forms comprised six male and six female faces; for each sex, there were two young adults (around 30 years old), two middle-aged adults (around 50 years old), and two elderly adults (around 70 years old). Before the experiments, we conducted a pilot study to ensure no differences existed between the parallel forms of the task. The task and its parallel form can be provided upon request.

The score was computed by deriving an accuracy percentage index, dividing by 12, and multiplying by 100 the correct association sum. The partial recall scores were computed in the same way, only considering the sum of face–name (NAME) and face–occupation (OCCUPATION) correctly recollected.

Each accuracy percentage index was computed for each task phase: immediate cued recall, delayed cued recall, and recognition for experiment 1; immediate cued recall, delayed cued recall (day 1), 24-hr delayed cued recall (day 2), 1 week cued recall and recognition (day 7) for experiment 2.

### The short-term memory binding test

This test was run between the FNAT immediate cued recall and the 15-min delayed cued recall. STMB is a recognition task relying on a change detection paradigm elaborated by Parra and colleagues (*Parra et al., 2014*). Participants were instructed to remember visual arrays of three black shapes (shape-only condition, *Figure 1—figure supplement 3A*) or colored shapes (shape-color binding condition, *Figure 1—figure supplement 3B*) presented for 2 s (study phase). After a 1-s delay where a blank screen is presented (retention interval), a display with the same or different items appears in new random locations on the screen (test phase). Participants were asked to press the '1' button on the keyboard if the items shown in the study and test phase were different (50% of the trials) or to press '2' if they were the same. A total of 32 randomized trials were presented for each condition. Conditions were counterbalanced across participants. Before starting the test, each participant underwent a perception trial where the two arrays of shapes were presented on the same screen, to exclude perceptual deficit and to train participants in the keyboard answers (*Figure 1—figure supplement 3*). The test was administered through E-prime 2.0 (Psychology Software Tools, Pittsburgh, USA), which recorded accuracy and reaction times (RTs). We derived two accuracy indexes, respectively, for the shape-only condition and the shape-color binding condition (total score of each condition/32 × 100).

### TMS–EEG data acquisition

Neurophysiological effects in cortical oscillations, excitability, and connectivity were assessed using single-pulse TMS during EEG recordings. During the TMS–EEG assessment, participants sat on a comfortable armchair in a soundproof room, were instructed to fixate a black cross on the wall, and wore in-ear plugs that continuously played a white noise to avoid possible auditory event-related

potential responses (*Rocchi et al., 2021*). TMS–EEG recordings were performed in a resting state to avoid contamination of cortical activity due to underlying memory tasks. The intensity of the white noise was adjusted individually by increasing the volume (always below 90 decibels) until the participant was sure that s/he could no longer hear the TMS-induced click. TMS for EEG recordings was the same stimulator as for the neuromodulation protocol. The stimulated areas were the PC, the area of interest, and the l-PPC, considered as a control area. The order of the stimulation was counterbalanced across participants. PC and l-PPC were both identified by the individual resting state structural MRI and fMRI (*Figure 1A*, see 'MRI data acquisition and preprocessing' paragraph). The coil position was constantly monitored using the Softaxic neuro-navigation system (E.M.S. Products, Bologna, Italy) and was differently oriented depending on the area of stimulation so that the direction of current flow in the most effective (second) phase was in the posterior–anterior direction. To target the PC, the coil was positioned with an orientation parallel to the midline, while to target the

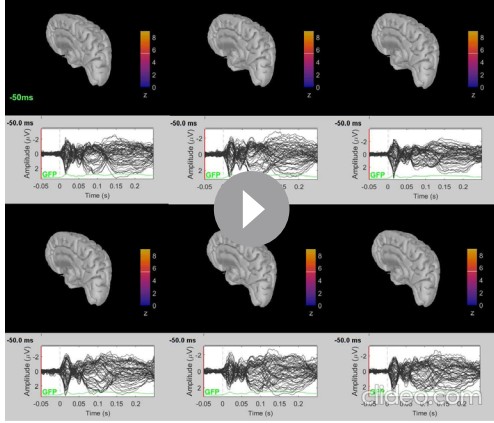

**Video 1.** TMS–EEG source reconstruction. The movie shows the source reconstruction of the TMS–EEG signal when targeting the precuneus. The three columns correspond to the recordings conducted before, just after, and 20 min from the non-invasive brain stimulation protocol (i.e., T0, T1, and T2). The first row corresponds to the iTBS + γtACS condition, while the second to the iTBS + sham-tACS condition.
https://elifesciences.org/articles/104220/figures#video1

l-PPC, the coil was positioned with an orientation of 15° from the midline (*Koch et al., 2018*). The single-pulse TMS intensity was set at 100% eSI, separately acquired for individual PC and l-PPC (see 'iTBS + γtACS neuromodulation protocol' paragraph) without the tACS electrode under the coil. Each TMS–EEG block consisted of 90 single pulses with a randomized inter-stimulus interval between 2 and 4 s. EEG was continuously recorded from 61 scalp sites positioned according to the 10–20 International System, using TMS-compatible Ag/AgCl pellet electrodes mounted on an elastic cap (BrainAmp; BrainProducts GmbH, Munich, Germany). Additional electrodes were used as a ground and reference: the ground electrode was positioned in Fpz, while the reference was positioned on the tip of the nose. EEG signals were digitized at a sampling rate of 5 kHz. Skin/electrode impedance was maintained below 5 kΩ.

## TMS–EEG preprocessing and analysis

### EEG preprocessing

TMS-EEG data were preprocessed offline with Brain Vision Analyzer (Brain Products GmbH, Munich, Germany). Data were segmented into epochs starting 1 s before and 1 s after the TMS pulse. The TMS pulse artifact was removed and replaced using cubic interpolation, from 1 ms before to 10 ms following the pulse. Afterward, data were down-sampled to 1000 Hz and band-pass filtered between 1 and 80 Hz (Butterworth zero-phase filters). A 50-Hz notch filter was applied to reduce noise from electrical sources. Then, all the epochs were visually inspected, and those with an excessively noisy EEG were excluded from the analysis. Independent component analysis (INFOMAX-ICA) was applied to the EEG signal to identify and remove components reflecting muscle activity, eye movements, blink-related activity, and residual TMS-related artifacts based on previously established criteria (*Casula et al., 2017*; *Hernandez-Pavon et al., 2023*). Finally, the signal was re-referenced to the average signal of all the electrodes.

### Source activation

To characterize cortical source activation at the individual anatomical level, each participant's MRI was uploaded to Brainstorm toolbox, segmented through CAT12 (*Gaser et al., 2022*), and normalized. TMS–EEG recordings were then uploaded for each subject and condition, adjusting the channel montage over the cortical surface extracted from the individual MRI. The head model was computed

using the symmetric boundary element method (symmetric BEM) running on Open MEEG (*Gramfort et al., 2010*; *Kybic et al., 2005*). Noise covariance was calculated referring to a baseline ranging from –500 to –1 ms relative to the TMS pulse and removing DC offset block by block. Source activity was computed adopting the minimum norm imaging method, the current density map was set as measure and a constrained dipole orientation was selected. The estimated sources were transformed into *z*-score points considering a baseline window of –100 to –1 ms and were then spatially smoothed (SurfStat, KJ, Worsley) with a full width at half maximum value set at 3 mm. Once source activity was calculated, we projected it to the default ICBM152 cortex surface to permit group visualization (*Video 1*).

### Cortical oscillations

The analysis of cortical oscillations was performed running a time/frequency decomposition based on Morlet wavelet (cycles: 5; frequency resolution: 1 Hz from 4 to 90 Hz; baseline correction –700; –300 ms) and computing the TRSP for theta, alpha, beta, and gamma bands (*Casula et al., 2022*; *Casula et al., 2018*). To analyze local oscillatory activity, we computed the average TRSP within the following clusters of electrodes: Pz, P1, P2, POz for PC; P3, P5, CP3, PO3 for l-PPC. For each time, condition, and stimulated area, we computed the corresponding cluster and averaged the TRSP values between 10 and 250 ms (i.e., the average time window of the spectral perturbation) in the following frequency bands: theta (4–7 Hz), alpha (8–13 Hz), beta (14–30 Hz), low-gamma (31–40 Hz), and high-gamma (41–90 Hz).

### Cortical excitability

Changes in cortical excitability were analyzed through the TEPs by importing the preprocessed EEG data in the Brainstorm toolbox (*Tadel et al., 2011*) running in a MATLAB environment (MathWorks Inc, Natick, MA). TEPs were computed by averaging all the time-locked EEG responses in each electrode from 100 ms before to 150 ms after the TMS pulse, with a baseline correction of 100 ms before the TMS pulse. TEP's amplitude was then computed as the mean activity within four time windows (W) based on an accurate visual inspection (*Casula et al., 2023*; *Koch et al., 2018*): W1 from 10 to 30 ms, W2 from 31 to 60 ms, W3 from 61 to 120 ms, and W4 from 121 to 145 ms after the TMS pulse.

## MRI data acquisition

MRI data were acquired (1) before experiments 1, 3, and 4, to identify and individualize the stimulation target and (2) before and after the neuromodulation protocol, to assess functional connectivity effects (*Figure 1*, panel C).

Imaging was acquired on a Siemens PRISMA scanner with a 64-channel head coil (Siemens, USA). During the first acquisition (1), both the structural MRI and fMRI were run, whereas in the following acquisition (2), only the fMRI was acquired.

The structural imaging was acquired using high-resolution T1-weighted (T1w) anatomical images obtained through a 3D-MPRAGE sequence (TR = 2500 ms, TE = 2 ms, TI = 1070 ms, flip angle = 8°, thickness = 1 mm, imaging matrix = 240 × 240), and a DTI sequence (TR = 3400 ms, TE = 80 ms; 121 directions, *b*-value=1000 s/mm$^2$, thickness = 1.79 mm, gap = 1.79 mm, flip angle = 90), only acquired before the experiments 3 and 4. The fMRI images were acquired using standard echo-planar blood oxygenation level-dependent (BOLD) imaging (TR = 800 ms, TE = 3 ms, flip angle = 52°, thickness = 2.4 mm, gap = 2.4 mm). Subjects were instructed not to focus their thoughts on any particular topic, not to cross their arms or legs, and to keep their eyes open.

For the individualization of the stimulation sites (*Figure 1*, panel A), the functional ROI representing the PC node of the DMN (for experiments 1, 3, and 4) and the l-PPC node of the fronto-parietal network (FPN) (for experiment 3) was derived from the Harvard-Oxford atlas available in CONN (*Whitfield-Gabrieli and Nieto-Castanon, 2012*). A seed-to-voxel correlation map was computed for each participant, thus obtaining a map of positively and negatively correlated voxels, respectively, representing the DMN and FPN. An investigator expert in MRI checked both rs-FC maps and structural MRI data (i.e., T1-weighted images) to identify individual hotspots based on ad hoc criteria. In particular, the stimulation sites were defined as the ones closest to the local maxima of the rs-FC cluster identified as DMN-PC and FPN-PPC, being on the top of a cortical gyrus and representing the shortest perpendicular path connecting the stimulating TMS coil on the scalp and the cortex. Based

on best judgment, the resulting set of coordinates was picked as the individual stimulation site. The individual set of coordinates was then transformed using a nonlinear transformation to reconstruct the targets in individual brain spaces. Lastly, to ensure a coherent intrasession and intersession stimulation (*Cocchi and Zalesky, 2018*; *Fitzgerald et al., 2009*), the individualized targets were marked in the subject's anatomical MRI and loaded into our neuronavigation system. Both personalized points were used for the TMS–EEG assessment, whereas only the personalized PC was used for the neuromodulation protocol in experiments 1, 3 and 4. fMRI data preprocessing.

SPM12 (Statistical Parametric Mapping), CONN, and MATLAB 2018a (MathWorks, MA, USA) software were used to preprocess datasets. The following preprocessing steps were applied to the BOLD images: discarding of the first three volumes to allow for steady-state magnetization and stabilization of participant status; slice timing; realigning to correct for head motion; co-registration to structural images; segmentation; nonlinear normalization to the Montreal Neurological Institute (MNI) template brain; voxel resampling to an isotropic 3 × 3 × 3 mm voxel size; smoothing with an isotropic Gaussian kernel (full-width at half maximum, 8 mm). Structural images were co-registered to the mean volume of functional images and segmented. Linear trends were removed to reduce the influence of the rising temperature of the MRI scanner and all functional volumes were band-pass filtered at 0.01 Hz < $f$ < 0.08 Hz to reduce low-frequency drifts. Finally, we regress out potential confounding signals, like physiological high-frequency respiratory and cardiac noise, from gray matter voxels' BOLD time course using the Compcorr algorithm (*Whitfield-Gabrieli and Nieto-Castanon, 2012*), to reduce artificial negative correlation and provide adequate filtering of the data.

## DTI preprocessing

The FMRIB Software Library (FSL) and DSI-studio were used to preprocess datasets. Binary masks were made and then the data were corrected for susceptibility artifacts using the TOPUP function. Volumes acquired with opposing phase-encoding directions were combined and then corrected for artifacts of motion and eddy-current distortion using EDDY. The corrected diffusion-weighted dataset and reoriented *b*-table were then imported to DSI Studio, registered (rigid body), and resampled to the space of T1. The accuracy of *b*-table orientation was examined by comparing fiber orientations with those of a population-averaged template (*Yeh et al., 2018*). The restricted diffusion was quantified using restricted diffusion imaging (*Yeh et al., 2017*). The diffusion data were reconstructed using generalized *q*-sampling imaging (*Yeh et al., 2010*) with a diffusion sampling length ratio of 1.25. The tensor metrics were calculated using DWI with a *b*-value lower than 1750 s/mm². A deterministic fiber tracking algorithm (*Yeh et al., 2013*) was used with augmented tracking strategies (*Yeh, 2020*) to improve reproducibility. The anatomy prior to a tractography atlas (*Yeh et al., 2018*) was used to map the bilateral MdLF with a distance tolerance of 16 mm in the ICBM152 space. The anisotropy threshold was randomly selected. The angular threshold was randomly selected from 15° to 90°. The step size was randomly selected from 0.5 to 1.5 voxels. Tracks with lengths shorter than 30 or longer than 300 mm were discarded. A total of 10000 tracts was calculated. Topology-informed pruning (*Yeh et al., 2019*) was applied to the tractography with 16 iterations to remove false connections. Finally, the MdLF FA was extracted for each subject.

## Statistical analysis

Memory performance and local oscillatory changes data were analyzed through SPSS v22 (IBM, Armonk, NY). Before undergoing parametric or non-parametric statistical procedures, the assumption of normal distribution of data residuals was assessed with the Shapiro–Wilk test. The assumption of sphericity was tested with the Mauchly test; if this test was significant, we used the Huynh–Feldt correction. The level of significance was set at $\alpha = 0.05$.

To assess memory performances in experiment 1, we run repeated-measures ANOVAs with stimulation condition as a within-subject factor (i.e., iTBS + γtACS; iTBS + sham-tACS; sham-iTBS + sham-tACS) for each dependent variable. In detail, a separate ANOVA was conducted for every FNAT measure (i.e., name, occupation, total) on each memory process (i.e., immediate recall, delayed recall, and recognition). Separate ANOVAs were also conducted for STBM accuracy and RTs, both for the shape and the binding condition. Post hoc comparisons were performed with paired *t*-tests corrected with the Bonferroni method.

To confirm the results obtained from experiment 1 and to investigate the long-lasting effect on long-term memory, we performed experiment 2, where we first ran a repeated-measures ANOVA with stimulation condition as a within-subject factor (i.e., iTBS + γtACS, iTBS + sham-tACS) for the immediate recall, as in experiment 1. Then, a two-way repeated-measures ANOVA was conducted for FNAT delayed recall performance, where stimulation condition (i.e., iTBS + γtACS, iTBS + sham-tACS) and time (i.e., days 1, 2, and 7) were considered as within-subject factors. Following the results obtained in this analysis (see 'results' paragraph), we conducted exploratory paired t-tests aimed at investigating the effect of stimulation conditions on each time point.

Since we do not have an a priori effect size for experiments 1 and 2, we performed a sensitivity power analysis to ensure that these experiments were able to detect the minimum effect size with 80% power and an alpha level of 0.05.

To assess cortical oscillation changes in experiment 3, we computed repeated-measures ANOVAs with stimulation condition (iTBS + γtACS, iTBS + sham-tACS) and time T1–T0, T2–T0 (ΔT1, ΔT2) as within factors for each frequency band. Also, in this case, paired t-tests were conducted to investigate the effect of stimulation conditions on each time point. A one-tailed hypothesis was considered, given the precise hypothesis of gamma increase deriving from our previous work (*Maiella et al., 2022*).

To assess cortical excitability effects, we used multiple paired t-tests. The analysis was run in the four time windows of the TEP waveform (i.e., 10–30, 21–60, 61–120, and 121–150 ms after TMS; see paragraph 'Cortical excitability' in 'TMS–EEG data acquisition and preprocessing') comparing T1 and T2 to T0 in each stimulation condition. To reduce the occurrence of type I errors, we used the Monte Carlo method, which computes the estimates of the significance probabilities from two surrogate distributions constructed by randomly permuting the two original conditions' data 3000 times. Moreover, p-values were corrected with the false discovery rate (FDR) method considering the number of electrodes and were considered significant when at least 10 successive t-tests reached the significance threshold (p < 0.05). The same statistical analysis was performed to assess changes in source activation. The time series extracted and analyzed from the ROIs (see paragraph 'source activation' in 'TMS–EEG data acquisition and preprocessing') ranged from −50 to 150 ms from the TMS pulse. As for excitability analysis, we avoided the occurrence of type I errors by implementing Monte Carlo permutation and considering a significant result when at least 10 successive t-tests reached the significance threshold (p < 0.05).

To test rs-FC modulation, we used a general linear model. The statistical analyses were carried out using the CONN (v.20b) toolbox and Matlab 2018b software (Mathworks, MA, USA) and were performed considering stimulation conditions (i.e., iTBS + γtACS and iTBS + sham-tACS) and time points (i.e., pre and post) as factors. In particular, we performed an ROI-to-ROI analysis focusing on six specific ROIs, based on our hypothesis: left and right PPC, medial prefrontal cortex, PC, left and right hippocampus, chosen based on the Harvard-Oxford (cortical and subcortical) atlas (https://fsl.fmrib.ox.ac.uk/fsl/docs/#/) (*Desikan et al., 2006*). Specifically, rs-FC changes were calculated by computing the Pearson correlation coefficient between the average time series extracted from each individual ROI. Multiple comparisons between ROIs were then performed using a two-sided contrast with a cluster level of p < 0.05 FDR-corrected. In addition, the significant nodes were considered as seeds for the voxel-wise analysis, specifically performed by comparing the pre- and post-iTBS + γtACS. Temporal correlations were calculated between these seeds and all other voxels in the brain. Results were computed by applying a cluster-level threshold of p < 0.05 FDR-corrected.

The sample size for experiments 3 and 4 was estimated based on a previous study using the same protocol (*Maiella et al., 2022*). Based on the effect size reported in this work ($\eta^2 = 0.291$), our power analysis estimated that a sample size of 14 patients would be necessary to obtain the same effect size with 80% power and an alpha level of 0.05.

To assess linear relationships between the white matter integrity and the functional connectivity outcomes, we perform a bivariate correlation between FA values extracted from the MdLF and the functional connectivity modulation induced by the stimulation assessed through the ROI-to-ROI analysis. Correlations were computed with Pearson's coefficient (two-tailed).

Data will be available at https://doi.org/10.5061/dryad.573n5tbn2.

## Acknowledgements

The authors would like to gratefully acknowledge Maria Stefania De Simone and Marta Rodini for their help in task design and the participants involved in the studies. The draft has been revised by Prof. Alessandro D'Ausilio and Luciano Fadiga. The study was funded by H2020 EUROPEAN COMMISSION Future and Emerging Technologies (FET) grant agreement No. 101017716 (GK) and the Italian Ministry of University and Research under the National Recovery and Resilience Plan [PE00000006 'A multiscale integrated approach to the study of the nervous system in health and disease' MNESYS (GK) and Fondi DM 502/2022 – PNRR MC42 Bando Giovani Ricercatori (EPC)].

## Additional information

### Competing interests

The authors declare that no competing interests exist.

### Funding

| Funder | Grant reference number | Author |
|---|---|---|
| H2020 Future and Emerging Technologies | 101017716 | Giacomo Koch |
| NextGenerationEU | PE00000006 | Giacomo Koch |
| NextGenerationEU | MC42 | Elias Paolo Casula |

The funders had no role in study design, data collection, and interpretation, or the decision to submit the work for publication.

### Author contributions

Ilaria Borghi, Conceptualization, Data curation, Formal analysis, Investigation, Visualization, Methodology, Writing - original draft, Project administration; Lucia Mencarelli, Data curation, Formal analysis, Visualization, Methodology, Writing – review and editing; Michele Maiella, Formal analysis, Investigation; Elias Paolo Casula, Conceptualization, Formal analysis, Methodology, Writing – review and editing; Matteo Ferraresi, Francesca Candeo, Elena Savastano, Investigation; Martina Assogna, Conceptualization, Visualization, Writing – review and editing; Sonia Bonnì, Conceptualization, Methodology, Writing – review and editing; Giacomo Koch, Conceptualization, Resources, Supervision, Funding acquisition, Visualization, Methodology, Project administration, Writing – review and editing

### Author ORCIDs

Ilaria Borghi ⓘ https://orcid.org/0000-0002-3058-3337
Lucia Mencarelli ⓘ https://orcid.org/0000-0002-4006-4629
Michele Maiella ⓘ https://orcid.org/0000-0003-4017-3769
Matteo Ferraresi ⓘ https://orcid.org/0009-0004-5074-6094
Giacomo Koch ⓘ https://orcid.org/0000-0001-6155-9439

### Ethics

All participants provided written informed consent approved by the Santa Lucia Foundation IRCCS ethical committee (CE/PROG.923) in accordance with the Declaration of Helsinki.

Reviewer #1 (Public review): https://doi.org/10.7554/eLife.104220.4.sa1
Reviewer #2 (Public review): https://doi.org/10.7554/eLife.104220.4.sa2
Reviewer #3 (Public review): https://doi.org/10.7554/eLife.104220.4.sa3
Author response https://doi.org/10.7554/eLife.104220.4.sa4

## Additional files

### Supplementary files

Supplementary file 1. Experiment 1 statistical details of the face–name associative task (FNAT) and

short-term memory binding test (STMB).

Supplementary file 2. Experiment 2 statistical details of the immediate and recognition face–name associative task (FNAT) and short-term memory binding test (STMB).

Supplementary file 3. Experiment 2 statistical details of the delayed face–name associative task (FNAT).

Supplementary file 4. Demographic characteristics.

Supplementary file 5. Subject-specific information about stimulation parameters and e-field calculations.

MDAR checklist

### Data availability

Data from all the experiments have been deposited in DRYAD (https://doi.org/10.5061/dryad. 573n5tbn2). Data related to subject-specific information about stimulation parameters and e-field calculations are included in the supplementary information.

The following dataset was generated:

| Author(s) | Year | Dataset title | Dataset URL | Database and Identifier |
|---|---|---|---|---|
| Borghi I, Mencarelli L, Maiella M, Casula EP, Ferraresi M, Savastano E, Assogna M, Candeo F, Bonnì S, Koch G | 2025 | Dual transcranial electromagnetic stimulation of the precuneus boosts human long-term memory | https://doi.org/ 10.5061/dryad. 573n5tbn2 | Dryad Digital Repository, 10.5061/dryad.573n5tbn2 |

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
