## [Editor Report · eLife Assessment]

This work presents **important** findings suggesting that a combination of transcranial stimulation approaches applied for a short period could improve memory performance. **Solid** methods and evidence, in line with current standards for non-invasive stimulation and recording, are included to broadly support the main findings. The results potentially have implications for non-invasive enhancement of cognitive functions.

---

## [Referee Report · Reviewer #1 (Public review)]

Summary:

The authors make a bold claim that a combination of repetitive transcranial magnetic stimulation (intermittent theta burst-iTBS) and transcranial alternating current stimulation (gamma tACS) causes slight improvements in memory in a face/name/profession task.

Strengths:

The idea of stimulating the human brain non-invasively is very attractive because, if it worked, it could lead to a host of interesting applications. The current study aims to evaluate one such exciting application.

---

## [Referee Report · Reviewer #2 (Public review)]

Summary:

The manuscript by Borghi and colleagues provides evidence that the combination of intermittent theta burst TMS stimulation and gamma transcranial alternating current stimulation (γtACS) targeting the precuneus increases long-term associative memory in healthy subjects compared to iTBS alone and sham conditions. Using a rich dataset of TMS-EEG and resting-state functional connectivity (rs-FC) maps and structural MRI data, the authors also provide evidence that dual stimulation increased gamma oscillations and functional connectivity between the precuneus and hippocampus. Enhanced memory performance was linked to increased gamma oscillatory activity and connectivity through white matter tracts.

Strengths:

The combination of personalized repetitive TMS (iTBS) and gamma tACS is a novel approach to targeting the precuneus, and thereby, connected memory-related regions to enhance long-term associative memory. The authors leverage an existing neural mechanism engaged in memory binding, theta-gamma coupling, by applying TMS at theta burst patterns and tACS at gamma frequencies to enhance gamma oscillations. The authors conducted a thorough study that suggests that simultaneous iTBS and gamma tACS could be a powerful approach for enhancing long-term associative memory. The paper was well-written, clear, and concise.

---

## [Referee Report · Reviewer #3 (Public review)]

Summary:

Borghi and colleagues present results from 4 experiments aimed at investigating the effects of dual γtACS and iTBS stimulation of the precuneus on behavioral and neural markers of memory formation. In their first experiment (n = 20), they find that a 3-minute offline (i.e., prior to task completion) stimulation that combines both techniques leads to superior memory recall performance in an associative memory task immediately after learning associations between pictures of faces, names, and occupation, as well as after a 15-minute delay, compared to iTBS alone (+ tACS sham) or no stimulation (sham for both iTBS and tACS). Performance in a second task probing short-term memory was unaffected by the stimulation condition. In a second experiment (n = 10), they show that these effects persist over 24 hours and up to a full week after initial stimulation. A third (n = 14) and fourth (n = 16) experiment were conducted to investigate neural effects of the stimulation protocol. The authors report that, once again, only combined iTBS and γtACS increases gamma oscillatory activity and neural excitability (as measured by concurrent TMS-EEG) specific to the stimulated area at the precuneus compared to a control region, as well as precuneus-hippocampus functional connectivity (measured by resting state MRI), which seemed to be associated with structural white matter integrity of the bilateral middle longitudinal fasciculus (measured by DTI).

Strengths:

Combining non-invasive brain stimulation techniques is a novel, potentially very powerful method to maximize the effects of these kinds of interventions that are usually well-tolerated and thus accepted by patients and healthy participants. It is also very impressive that the stimulation-induced improvements in memory performance resulted from a short (3 min) intervention protocol. If the effects reported here turn out to be as clinically meaningful and generalizable across populations as implied, this approach could represent a promising avenue for treatment of impaired memory functions in many conditions.

Methodologically, this study is expertly done! I don't see any serious issues with the technical setup in any of the experiments. It is also very commendable that the authors conceptually replicated the behavioral effects of experiment 1 in experiment 2 and then conducted two additional experiments to probe the neural mechanisms associated with these effects. This certainly increases the value of the study and the confidence in the results considerably.

The authors used a within-subject approach in their experiments, which increases statistical power and allows for stronger inferences about the tested effects. They also used to individualize stimulation locations and intensities, which should further optimize the signal-to-noise ratio.

---

## [Author Response]

The following is the authors’ response to the previous reviews

**Reviewer #1 (Public review):**
Summary:The authors make a bold claim that a combination of repetitive transcranial magnetic stimulation (intermittent theta burst-iTBS) and transcranial alternating current stimulation (gamma tACS) causes slight improvements in memory in a face/name/profession task.Strengths:The idea of stimulating the human brain non-invasively is very attractive because, if it worked, it could lead to a host of interesting applications. The current study aims to evaluate one such exciting application.Weaknesses:(1) The title refers to the "precuneus-hippocampus" network. A clear definition of what is meant by this terminology is lacking. More importantly, mechanistic evidence that the precuneus and the hippocampus are involved in the potential effects of stimulation remains unconvincing.

Thank you for the observation. We believe that the evidence collected supports our state relative to the stimulation of the precuneus and the involvement of the hippocampus. In particular, given the existing evidence on TMS methodology and precuneus non-invasive stimulation (see Koch et al., Brain, 2022, Koch et al., Alzheimer's research & therapy, 2025), the computation of the biophysical model with the E-field we produced (see Biophysical modeling and E-field calculation section in the supplementary information), together with the individual identification of the precuneus through the RM (see iTBS+γtACS neuromodulation protocol and MRI data acquisition in the main text), we can reasonably assume that the individually identified PC was stimulated.

As we acknowledged in the Limitations section, we cannot entirely rule out the possibility that our results might also reflect stimulation of more superficial parietal regions adjacent to the precuneus. Nor do we provide direct evidence of microscopic changes in the precuneus following stimulation. However, the results we provide in terms of changes in precuneus oscillatory activity and precuneus-hippocampi connectivity sustain both our thesis of the precuneus stimulation and of hippocampi involvement in the stimulation effects.

Despite this consideration, we agree on the fact that a clear definition of what is meant by the terminology “precuneus-hippocampus network” is lacking. Moreover, since our data and previous evidence sustain the notion of PC stimulation, while this study does not produce direct evidence of the hippocampi stimulation - but only of the effect of the neuromodulation protocol on its connection with the precuneus, we soften the claim in the title. We remove the mention of the precuneus-hippocampus network so that the modified title will be as follows: “Dual transcranial electromagnetic stimulation of the precuneus boosts human long-term memory.”

(2) The question of the extent to which the stimulation approach and the stimulation parameters used in these experiments causes specific and functionally relevant neural effects remains open. Invasive recordings that could address this question remain out of the scope of this non-invasive study. The authors conducted scalp EEG experiments in an attempt to address this question using non-invasive methods. However, the results shown in Fig. 3 are unclear. The results are inconsistently reported in units of microvolts squared in some panels (3A, 3B) and in units of microvolts in other panels (3C). Also, there is insufficient consideration of potential contamination by signal components reflecting eye movements, other muscle artifacts, or another volume-conducted signal reflecting aggregate activity inside the brain.

As you correctly noted, Figure 3 presents results obtained from the TMS–EEG recordings. However, there is no inconsistency regarding the measurement units, as we are referring to two distinct indices: one in the frequency domain—oscillatory power shown in Figures 3A and 3B, expressed in microvolts squared (μV^²^)—and one in the time domain—the TMS-evoked potential shown in Figure 3C, expressed in microvolts (μV).

Regarding the concern about artifacts, this is an important issue on which our group has a strong expertise, having published well-established, highly cited procedures on how to record and clean TMS-EEG signals (e.g., Casula et al., Clinical Neurophysiology, 2017; Rocchi et al., Brain Stimulation, 2021). In the current study, we adopted a well-established and rigorous approach for both data acquisition and preprocessing. This ensured that the recorded TMS–EEG signals were not contaminated by physiological or electrical artifacts.

As regards the recording procedure, all participants were instructed to fixate on a black cross to minimize eye movements. To avoid auditory-related components caused by the TMS click, we adopted an ad-hoc procedure optimized for TMS-EEG recordings (Rocchi et al., Brain Stimulation, 2021). First, participants were given earphones that continuously played an ad-hoc masking noise composed of white noise mixed with specific time-varying frequencies of the TMS click (Rocchi et al., Brain Stimulation, 2021). The masking noise volume was adjusted to ensure that participants could not detect the TMS click, or as much as tolerated (always below 90 dB). To further reduce the impact of the TMS click on the EEG signal, we placed ear defenders (SNR=30) on top of the earphones. Please see TMS–EEG data acquisition section in the main text.

As regards the offline cleaning process, we applied Independent Component Analysis (INFOMAX-ICA) to the EEG data to identify and remove components associated with muscle activity, eye movements, blinking, and residual TMS-related artifacts, in line with the most recent guidelines on TMS–EEG preprocessing (Hernandez-Pavon et al., Brain Stimulation, 2023). Specifically, for TMS-related muscle artefacts, we strictly followed the criteria based on their scalp topography, spectral content, timing, and amplitude, which we published in a paper focused on this topic (Casula et al., Clinical Neurophysiology, 2017). We add this detail in the TMS–EEG preprocessing and analysis section in the supplementary information (lines 119-120).

(3) Figure 3 indicates "Precuneus oscillatory activity ...", but evidence that the activity presented reflects precuneus activity is lacking. The maps shown at the bottom of Figure 3C suggest that the EEG signals recorded with scalp EEG reflect activity generated across a wide spatial range, with a peak encompassing at least tens of centimeters. Thus, evidence that effects specifically reflect precuneus activity, as the paper's title and text throughout the manuscript suggest, is lacking.

We believe there may have been a misunderstanding. As indicated in the figure caption, panels A and B represent oscillatory activity, whereas panel C displays the TMS-evoked potentials (TEPs). Therefore, the topographical maps mentioned (i.e., those in panel C) did not refer to oscillatory activity, but to differences in TEP amplitude. Specifically, the topographies shown in Figure 3C illustrate statistically significant differences in TEP amplitudes between post-stimulation time points (T1—immediately after stimulation, and T2—20 minutes after stimulation) and the pre-stimulation baseline (T0).

In this figure, we focused our analysis on a cluster of electrodes overlying the individually identified precuneus, capturing EEG responses to single TMS pulses delivered to that target. This approach, widely used in previous literature (e.g., Koch et al., NeuroImage, 2018; Casula et al., Annals of Neurology, 2022; Koch et al., Brain, 2022; Maiella et al., Clinical Neurophysiology, 2024; Koch et al., Alzheimer’s Research & Therapy, 2025), supports the interpretation that the observed responses reflect precuneus-related activity. Furthermore, the wide spatial range change you mention proved to be statistically different only when conducting the TMS-EEG over the precuneus (i.e., administering the TMS single pulse over the precuneus) and not when performing it over the left parietal cortex. We modified the discussion section in the main text to make it more clear (lines 196-199).

“Moreover, we observed specific cortical changes in the posteromedial parietal areas, as evidenced by the whole-brain analysis conducted on TMS-EEG data when performed over the precuneus and the absence of effect when TMS-EEG was performed on the lateral posterior parietal cortex used as a control condition.”

That said, we do not state that the effects observed specifically reflect the precuneus activity; indeed, we think the effect of the stimulation is broader, as discussed in the Discussion section. We rather sustain, in line with the literature (Koch et al., Neuroimage 2018; Koch et al., Brain, 2022; Koch et al., Alzheimer's research & therapy, 2025), the idea that the effects observed are a consequence of the precuneus stimulation by the dual stimulation.

(4) The paper as currently presented (e.g., Figure 3) also lacks rigorous evidence of relevant oscillatory activity. Prior to filtering EEG signals in a particular frequency band, clear evidence of oscillations in the frequency band of interest should be shown (e.g., demonstration of a clear peak that emerges naturally in the frequency range of interest when spectral analysis is applied to "raw" signals). The authors claim that gamma oscillations change because of the stimulation, but a clear peak in the gamma range prior to stimulation is not apparent in the data as currently presented. Thus, the extent to which spectral measurements during stimulation reflect physiological gamma oscillations remains unclear.

If we understand correctly, your concern relates to the lack of a clear gamma peak before neuromodulation, which may suggest uncertainty about the observed changes in gamma oscillatory activity. Is that correct?

First, it is important to underline that the natural frequency typically observed in the precuneus falls within the beta range, not the gamma range (see Rosanova et al., Journal of Neuroscience, 2009; Casula et al., Annals of Neurology, 2022). This explains why a prominent gamma peak is not expected at baseline (T0).

Differently, our neuromodulatory protocol was specifically aimed at boosting gamma oscillatory activity given its well-established role in learning and memory processes (Griffiths & Jensen, Trends in Neurosciences, 2023). Thus, to assess the effect of the neuromodulatory protocol, we compared the oscillatory activity before (T0) and after stimulation (T1 and T2), which showed a clear increase in the gamma band. This effect is visible in the raw oscillatory power plot and is most clearly represented in Figure 3B, where the gamma band emerged as the only frequency range showing significant changes across time points.

(5) Concerns remain regarding the rigor of statistical analyses in the revised manuscript (see also point 8 below). Figure 3B shows an undefined statistical test with p<0.05. The statistical test that was used is not explained. Also, a description of how corrections for multiple comparisons were made is missing. Figures 3A and 3C are not accompanied by statistics, making the results difficult to interpret. For Figure 4C, a claim was made based on a significant p-value for one statistical test and a non-significant p-value in another test. This is a common statistical mistake (see Figure 1 and accompanying discussion in Makin and Orban de Xivry (2019) Science Forum: Ten common statistical mistakes to watch out for when writing or reviewing a manuscript. eLife 8:e48175).

All statistical tests are described in the Statistical Analysis section of the main text. Specifically, to assess cortical oscillation changes in Experiment 3, we conducted repeated-measures ANOVAs with stimulation condition (iTBS+γtACS vs. iTBS+sham-tACS) and time (ΔT1 = T1–T0; ΔT2 = T2–T0) as within-subject factors, for each frequency band. To further explore the effects of stimulation at each time point, we performed paired t-tests with Bonferroni correction for multiple comparisons. A one-tailed hypothesis was adopted, based on our a priori prediction of gamma-band increase derived from previous work (Maiella et al., 2022).

Please note that Figures 3A and 3C are purely descriptive and are therefore not accompanied by statistical tests. Figure 3A shows the full spectral profile across frequencies and conditions, while statistical significance for these data is reported in Figure 3B. Similarly, the upper part of Figure 3C displays the TMS-evoked potential (TEP) in the precuneus, while the statistical comparison of TEP amplitudes across time points is shown in the lower part of Figure 3C.

Regarding Figure 4C and the article you cited, are you referring to the error described as “Interpreting comparisons between two effects without directly comparing them”? If we understand correctly, this refers to the mistake of inferring an effect by observing that a significant result occurs in one condition or group, while the corresponding result in another condition or group is not significant, without directly testing the difference between them.

In the case of Experiment 4, which investigates fMRI effects and is illustrated in Figure 4, we employed a general linear model that explicitly modeled both conditions and time points, allowing for a direct statistical comparison. Therefore, the connectivity effect reported does not fall into the category of the error you mentioned.

Importantly, Figure 4C does not depict the effect of the neuromodulatory protocol itself. Rather, its purpose is to show that, within the real stimulation condition, there is a correlation between the observed effect and the integrity of the bilateral Middle Longitudinal Fasciculus. No conclusions or assumptions were made based on the absence of a significant correlation in the sham condition. However, since it was an exploratory analysis, we decided to soften our claims relative to the neural mechanism in the discussion section of the main text (lines 241-246).

(6) In the second question posed in the original review, I highlighted that it was unclear how such stimulation would produce memory enhancement. The authors replied that, in the absence of mechanisms, there are many other studies that suffer from the same problem. This raises the question of placebo effects. The paper does not sufficiently address or discuss the possibility that any potential stimulation effects may reflect placebo effects.

We agree with the reviewer on the potential role of a placebo effect in our study. For this reason, our experimental study had several stimulation conditions, including a placebo condition, which corresponded to the sham iTBS-sham tACS condition, which did not produce any effect.

(7) The third major concern in the original review was the lack of evidence for a mechanism that is specific to the precuneus. Evidence for specific involvement of the precuneus remains lacking in the revised manuscript. The authors state: "the non-invasive stimulation protocol was applied to an individually identified precuneus for each participant". However, the meaning of this statement is unclear. Specifically, it is unclear how the authors know that they are specifically targeting the precuneus. Without directly recording from the precuneus and directly demonstrating effects, which is outside of the scope of the study, specific involvement of the precuneus seems speculative. Also, it does not seem as though a figure was included in the paper to show how the stimulation protocol specifically targets the precuneus. In their response to the original reviews, the authors state that posterior medial parietal areas are the only regions that show significant differences following the stimulation, but they did not cite a specific figure, or statistics reported in the text, that show this. In any event, posterior medial parietal areas encompass a wide area of the brain, so this would still not provide evidence for an effect specifically involving the precuneus.

We respectfully disagree with the claim that targeting the precuneus in our study is speculative. The statement that “without directly recording from the precuneus and directly demonstrating effects, which is outside the scope of the study, specific involvement of the precuneus seems speculative” would, by that logic, implicitly call into question a large body of cognitive neuroscience research employing non-invasive techniques such as EEG and fMRI.

Our methodological approach—combining MRI-guided stimulation, biophysical modeling, and TMS–EEG—is well established and widely used for targeting and studying the role of specific cortical regions, including the precuneus (e.g., Wang et al., Science, 2014; Koch et al., NeuroImage, 2018; Casula et al., Annals of Neurology, 2022, 2023; Koch et al., Brain, 2022; Maiella et al., Clinical Neurophysiology, 2024; Koch et al., Alzheimer’s Research & Therapy, 2025).

In line with previously published protocols (Santarnecchi et al., Human Brain Mapping, 2018; Özdemir et al., PNAS, 2020; Mantovani et al., Journal of Psychiatric Research, 2021), we identified individual targets (i.e., the precuneus) for each participant based on structural and resting-state functional MRI data (see MRI Data Acquisition and Preprocessing section in the main text). This target was then accurately localized using MRI-guided stereotaxic neuronavigation, ensuring reproducible and anatomically precise stimulation across subjects.

Finally, concerning the last comment about the lack of figures/statistics showing how the stimulation protocol targets the precuneus and the specificity of the effect observed, we would like to let the focus go over:

Figure 3 in the main text, where we show the results of the TME-EEG over the posterior medial parietal areas;

Figure S1 in the supplementary information, which shows with the e-fied simulation how the stimulation protocol targets the brain;

the Precuneus iTBS+γtACS increases gamma oscillatory activity section in the main text results, where we report the results of the statistical analysis of the TMS-EEG conducted over the precuneus and the left posterior parietal cortex, used as a control condition to test for the specificity of the neuromodulation protocol.

(8) Regarding chance levels, it is unfortunate that the authors cannot quantify what chance levels are in the immediate and delayed recall conditions. This makes interpretation of the results challenging. In the immediate and delayed conditions, the authors state that the chance level is 33%. It would be useful to mark this in the figures. If I understand correctly, chance is 33% in Fig. 2A. If this is the case and if I am interpreting the figure correctly:Gray bars for the sham condition appear to be below chance (~20-25%). Why is this condition associated with an accuracy level that is lower than chance?Cyan bars and red bars do not appear to be significantly different from chance (i.e., 33%), with red slightly higher than cyan. What statistic was performed to obtain the level of significance indicated in the figure? The highest average value for the red condition appears to be around 35%. More details are needed to fully explain this figure and to support the claims associated with this figure.

The immediate and recall conditions you mention correspond to a free recall task. In this case, the notion of a fixed "chance level" is not straightforward as it would be in recognition or forced-choice paradigms, which is why we did not quantify it at first. I will now try to explain this extensively.

Unlike multiple-choice tasks, where participants select the answer from a limited set of alternatives and the probability of a correct response by chance can be precisely quantified (e.g., 33% in a 3-alternative forced choice), free recall involves the spontaneous retrieval of items from memory without external cues or predefined options. As such, the response range in free recall is essentially unconstrained, encompassing the entire vocabulary of the participant.

Because of this open-ended nature, the probability of correctly recalling a studied item purely by chance is exceedingly low and could be approximated to zero. Also, in our task, participants had to correctly recollect both name and occupation, doubling the possibility of the answers.

This assumption is further supported by the fact that random guesses in free recall are unlikely to match any of the studied items, given the vast number of possible alternatives. As a result, performance above zero can be reasonably interpreted as reflecting genuine memory retrieval, rather than random guessing.

As regards statistics, repeated-measures ANOVAs with stimulation condition as a within-subject factor (i.e., iTBS+γtACS; iTBS+sham-tACS; sham-iTBS+sham-tACS) for each dependent variable (see statistical analysis section in main text).

(9) In the revised version of the paper, the authors did not address concerns associated with the block design (please see question 4d in the original review).

We are sorry for the misunderstanding. We did not address your concerns related to block design since it does not apply to our study. As reported in the paper you mentioned in the original review, block design involves data collection performed in response to different stimuli of a given class presented in succession. If this is the case, it does not correspond to our experimental design since both TMS-EEG and fMRI were conducted in the resting state (i.e., without the presentation of stimuli) on different days according to the different randomized stimulation conditions.

In sum, this study presents an admirable aspirational goal, the notion that a non-invasive stimulation protocol could modulate activity in specific brain regions to enhance memory. However, the evidence presented at the behavioral level and at the mechanistic level (e.g. the putative involvement of specific brain regions) remains unconvincing.

We hope our response will be carefully considered, fostering a constructive exchange and leading to a reassessment of your evaluation.

**Reviewer #2 (Public review):**
Summary:The manuscript by Borghi and colleagues provides evidence that the combination of intermittent theta burst TMS stimulation and gamma transcranial alternating current stimulation (γtACS) targeting the precuneus increases long-term associative memory in healthy subjects compared to iTBS alone and sham conditions. Using a rich dataset of TMS-EEG and resting-state functional connectivity (rs-FC) maps and structural MRI data, the authors also provide evidence that dual stimulation increased gamma oscillations and functional connectivity between the precuneus and hippocampus. Enhanced memory performance was linked to increased gamma oscillatory activity and connectivity through white matter tracts.Strengths:The combination of personalized repetitive TMS (iTBS) and gamma tACS is a novel approach to targeting the precuneus, and thereby, connected memory-related regions to enhance long-term associative memory. The authors leverage an existing neural mechanism engaged in memory binding, theta-gamma coupling, by applying TMS at theta burst patterns and tACS at gamma frequencies to enhance gamma oscillations. The authors conducted a thorough study that suggests that simultaneous iTBS and gamma tACS could be a powerful approach for enhancing long-term associative memory. The paper was well-written, clear, and concise.Comments on Revision:I thank the authors for their thoughtful responses to my first review and their inclusion of more detailed methodological discussion of their rationale for the stimulation protocol conditions and timing. Regarding the apparent difference in connectivity at baseline between conditions, the explanation that this is due to intrinsic dynamics, state, or noise implies the baseline is reflecting transient changes in dynamics rather than a true or stable baseline. Based on this, it looks like iTBS solely is significantly greater than the baseline before the iTBS and γtACS condition but maybe not that much lower than post-stimulation period for iTBS and γtACS. A longer baseline period should be used to ensure transient states are not driving baseline levels such that these endogenous fluctuations would average out. This also raises questions about whether the effect of iTBS and γtACS or iTBS alone are dependent on the intrinsic state at the time when stimulation begins. Their additional clarification of memory scoring is helpful but also reveals that the effect of dual iTBS+γtACS specifically on the association between faces and names is just significant. This modest increase in associative memory should be taken into consideration when interpreting these findings.

We thank the reviewer for the feedback. We fully agree that considering baseline dynamics is critical when assessing the neurophysiological and connectivity effects of stimulation protocols.

In Experiments 3 and 4, baseline measurements were specifically included in our design to account for the possibility that intrinsic dynamics, state, or noise could influence the observed effects of neuromodulation. Indeed, if we had compared only post-stimulation connectivity between the real and sham conditions, the effects might have appeared larger. The inclusion of baseline measurements allows us to contextualize and better isolate the neuromodulatory impact by controlling such endogenous fluctuations. Importantly, the fMRI connectivity measurements, which comprise the baseline, are derived from 10-minute BOLD signal acquisitions, which help mitigate the influence of transient fluctuations and provide a quite stable estimate of intrinsic connectivity.

Moreover, regarding the possibility that stimulation effects may depend on the intrinsic state at stimulation onset, we hypothesize that gamma-frequency entrainment induced by tACS could reduce the variability of intrinsic dynamics, promoting a more stable neural state that is favorable for the induction of long-term plasticity.

As regards the memory scoring, we would like to clarify that the significant improvement observed in the dual iTBS+γtACS condition does not pertain solely to the face–name association. Rather, it concerns the more demanding task of recalling the association between face, name, and occupation. While we agree that the observed effect could be considered modest, it is worth noting that it follows from only 3 minutes of stimulation.

**Reviewer #3 (Public review):**
Summary:Borghi and colleagues present results from 4 experiments aimed at investigating the effects of dual γtACS and iTBS stimulation of the precuneus on behavioral and neural markers of memory formation. In their first experiment (n = 20), they find that a 3-minute offline (i.e., prior to task completion) stimulation that combines both techniques leads to superior memory recall performance in an associative memory task immediately after learning associations between pictures of faces, names, and occupation, as well as after a 15-minute delay, compared to iTBS alone (+ tACS sham) or no stimulation (sham for both iTBS and tACS). Performance in a second task probing short-term memory was unaffected by the stimulation condition. In a second experiment (n = 10), they show that these effects persist over 24 hours and up to a full week after initial stimulation. A third (n = 14) and fourth (n = 16) experiment were conducted to investigate neural effects of the stimulation protocol. The authors report that, once again, only combined iTBS and γtACS increases gamma oscillatory activity and neural excitability (as measured by concurrent TMS-EEG) specific to the stimulated area at the precuneus compared to a control region, as well as precuneus-hippocampus functional connectivity (measured by resting state MRI), which seemed to be associated with structural white matter integrity of the bilateral middle longitudinal fasciculus (measured by DTI).Strengths:Combining non-invasive brain stimulation techniques is a novel, potentially very powerful method to maximize the effects of these kinds of interventions that are usually well-tolerated and thus accepted by patients and healthy participants. It is also very impressive that the stimulation-induced improvements in memory performance resulted from a short (3 min) intervention protocol. If the effects reported here turn out to be as clinically meaningful and generalizable across populations as implied, this approach could represent a promising avenue for treatment of impaired memory functions in many conditions.Methodologically, this study is expertly done! I don't see any serious issues with the technical setup in any of the experiments. It is also very commendable that the authors conceptually replicated the behavioral effects of experiment 1 in experiment 2 and then conducted two additional experiments to probe the neural mechanisms associated with these effects. This certainly increases the value of the study and the confidence in the results considerably.The authors used a within-subject approach in their experiments, which increases statistical power and allows for stronger inferences about the tested effects. They also used to individualize stimulation locations and intensities, which should further optimize the signal-to-noise ratio.Weaknesses:I think one of the major weaknesses of this study is the overall low sample size in all of the experiments (between n = 10 and n = 20). This is, as I mentioned when discussing the strengths of the study, partly mitigated by the within-subject design and individualized stimulation parameters. The authors mention that they performed a power analysis but this analysis seemed to be based on electrophysiological readouts similar to those obtained in experiment 3. It is thus unclear whether the other experiments were sufficiently powered to reliably detect the behavioral effects of interest. In the revised manuscript, the authors provide post-hoc sensitivity analyses that help contextualize the strength of the findings.While the authors went to great lengths trying to probe the neural changes likely associated with the memory improvement after stimulation, it is impossible from their data to causally relate the findings from experiments 3 and 4 to the behavioral effects in experiments 1 and 2. This is acknowledged by the authors and there are good methodological reasons for why TMS-EEG and fMRI had to be collected in separate experiments, but readers should keep in mind that this limits inferences about how exactly dual iTBS and γtACS of the precuneus modulate learning and memory.

We thank the reviewer for the feedback.

**Reviewer #1 (Recommendations for the authors):**
I suggest:(1) Removing all mechanistic claims about the precuneus and hippocampus.

We soften our claims about the precuneus-hippocampus network.

(2) Repeating and focusing on the behavioral experiments with a much larger number of images and stronger statistical power to try to demonstrate a compelling behavioral correlate of the proposed stimulation protocol.

We clarified the misunderstanding relative to the chance level of the behavioral experiments raised by the reviewer.

**Reviewer #2 (Recommendations for the authors):**
Use longer baseline to establish stable gamma level for comparisons in Figure 3

If we understand correctly, you propose to increase the baseline to establish the gamma oscillatory activity as expressed in Figure 3 (showing the results of experiment 3). Is that right? In the figure, you see a baseline of -100; 0ms, which we use for a merely graphical reason, since no activity is usually observable before the TMS pulse. However, to establish the level of gamma, we used a larger baseline correction ranging from -700 ms to -300 ms (i.e., 400ms). We added this important information in the cortical oscillation section of the supplementary information (lines 134-135).

**Reviewer #3 (Recommendations for the authors):**
I think that the authors did a great job responding to the concerns raised by the reviewers. All of my own comments have been satisfactorily addressed. I will update my public review to be more concise, so that it only includes the overall assessment of the manuscript, including the strengths and weaknesses, but without the requests for clarification. Strengths and weaknesses remain largely the same, as the authors did not conduct additional experiments.

Thank you.